# Robust vestibular self-motion signals in macaque posterior cingulate region

Bingyu Liu[1,2], Qingyang Tian[1,2], Yong Gu[1,2]*

[1]CAS Center for Excellence in Brain Science and Intelligence Technology, Key Laboratory of Primate Neurobiology, Institute of Neuroscience, Chinese Academy of Sciences, Shanghai, China; [2]University of Chinese Academy of Sciences, Beijing, China

**Abstract** Self-motion signals, distributed ubiquitously across parietal-temporal lobes, propagate to limbic hippocampal system for vector-based navigation via hubs including posterior cingulate cortex (PCC) and retrosplenial cortex (RSC). Although numerous studies have indicated posterior cingulate areas are involved in spatial tasks, it is unclear how their neurons represent self-motion signals. Providing translation and rotation stimuli to macaques on a 6-degree-of-freedom motion platform, we discovered robust vestibular responses in PCC. A combined three-dimensional spatiotemporal model captured data well and revealed multiple temporal components including velocity, acceleration, jerk, and position. Compared to PCC, RSC contained moderate vestibular temporal modulations and lacked significant spatial tuning. Visual self-motion signals were much weaker in both regions compared to the vestibular signals. We conclude that macaque posterior cingulate region carries vestibular-dominant self-motion signals with plentiful temporal components that could be useful for path integration.

## Introduction

Navigation is a fundamental and indispensable ability for creatures to survive and live in the world, such as in foraging and exploration. In cognitive studies, landmark-based and vector-based navigation are two commonly used strategies for spatial tasks (*Gallistel, 1990*). Landmarks are typically strong visual cues for helping localize and calibrate self-positions, yet this strategy is easily affected by environments such as open fields of deserts and grasslands, or woods without clear and reliable markers. Vector-based navigation, or path integration on the other hand, involves momentary self-motion cues from multiple sensory channels such as optic flow and vestibular ones for updating and accumulating one's heading direction and location (*Etienne and Jeffery, 2004*; *Valerio and Taube, 2012*). For example, linear and angular vestibular signals originating from peripheral otolith and semicircular canal organs are thought to provide critical inputs to head direction cells (HD cells), place cells or grid cells in the hippocampal-entorhinal navigation system (*Knierim et al., 1998*; *Stackman et al., 2002*; *Clark and Taube, 2012*; *Hitier et al., 2014*; *Jacob et al., 2014*; *Winter et al., 2015*).

To date, robust vestibular and visual signals related with self-motion have been discovered in numerous cortical areas including the dorsal part of medial superior temporal sulcus (MSTd) (*Duffy and Wurtz, 1997*; *Page and Duffy, 2003*; *Gu et al., 2006*), the ventral intraparietal area (VIP) (*Bremmer et al., 1999*; *Bremmer et al., 2002*; *Zhang and Britten, 2004*; *Chen et al., 2011a*; *Chen et al., 2011b*), the visual posterior sylvian area (VPS) (*Chen et al., 2011c*), parietal insular vestibular cortex (PIVC) (*Grüsser et al., 1990a*, *Grüsser et al., 1990b*; *Chen et al., 2010*; *Chen et al., 2011b*), the smooth eye movement of the frontal eye field (FEFsem) (*Gu et al., 2016*), and 7a (*Avila et al., 2019*), composing a network (*Guldin and Grüsser, 1998*; *Britten, 2008*; *Cheng and Gu, 2018*; *Gu, 2018*). However, most of these areas (except for 7a) have not shown direct

*For correspondence:
guyong@ion.ac.cn

Competing interests: The authors declare that no competing interests exist.

connections with the hippocampal navigation system. What regions may bridge the cortical self-motion system and the limbic hippocampal navigation system? One hypothesis is that PCC and RSC may be one of these hubs (*Rushworth et al., 2006*; *Vincent et al., 2010*; *Kravitz et al., 2011*). On one hand, there is evidence that PCC and RSC directly project to entorhinal and parahippocampal cortices (*Rosene and Van Hoesen, 1977*; *Vogt et al., 1979*; *Insausti et al., 1987*; *Vogt et al., 1987*; *Morecraft et al., 1989*). On the other hand, these areas are also heavily connected with cortical areas including area 7a (*Pandya et al., 1981*; *Vogt et al., 1987*; *Musil and Olson, 1988*; *Cavada and Goldman-Rakic, 1989*; *Olson and Musil, 1992*; *Shinder and Taube, 2010*), MSTd (*Akbarian et al., 1994*; *De Castro et al., 2020*), FEF (*De Castro et al., 2020*), PIVC, VPS and 3a (*Guldin and Grüsser, 1998*; *De Castro et al., 2020*). Thus, posterior cingulate region may be one important hub mediating the propagation of self-motion signals from parietal-temporal cortices to the hippocampal-entorhinal system.

Supporting this notion, numerous studies have provided evidence showing that posterior cingulate region is involved in spatial navigation. For example, navigation related neurons in PCC are reported when monkeys navigate in a virtual environment (*Sato et al., 2006*; *Sato et al., 2010*). Lesion studies confirmed that PCC in monkeys (*Sutherland et al., 1988*; *Murray et al., 1989*) and the homologues area in rodents were actively involved in spatial memory tasks (*Sutherland et al., 1988*; *Cooper et al., 2001*; *Cooper and Mizumori, 2001*; *Whishaw et al., 2001*). Using functional MRI with galvanic or caloric stimulation, vestibular signals were discovered in human PCC (*Smith et al., 2012*; *Schindler and Bartels, 2018*), and visual signals were reported in human and macaque PCC (*Wall and Smith, 2008*; *Fischer et al., 2012*; *Cottereau et al., 2017*; *Smith et al., 2017*; *Schindler and Bartels, 2018*). Similarly, RSC has also been implicated to be involved in navigation context for a long time, mainly from studies in rodents and human. For example, HD cells have been reported in rat RSC (*Cullen and Taube, 2017*), and inactivation of RSC disrupts HD cell and place cell activities in hippocampal circuit (*Cooper and Mizumori, 2001*; *Clark et al., 2010*). In human, RSC is active in virtual navigational tasks (see reviews by *Maguire, 2001*; *Spiers and Maguire, 2007*; *Vann et al., 2009*). In addition, RSC is directly involved in spatial coordinate transformation (*Iaria et al., 2007*; *Epstein, 2008*; *Hashimoto and Nakano, 2014*), which may be crucial for transforming egocentric self-motion signals to allocentric navigational signals in the hippocampal system (*Byrne et al., 2007*; *Wang et al., 2018*).

Although signals related with spatial perception have been previously indicated in posterior cingulate region, previous studies often include too complex cognitive factors or do not apply techniques with enough spatiotemporal resolution for pinning down properties of neuronal signals in these areas. Hence in our current study, we systematically examined self-motion related signals in macaque PCC and RSC on a virtual reality system with a 6-degree-of-freedom motion platform. We recorded single unit activities in both subregions in the posterior region of the cingulate cortex under linear translation and angular rotation conditions. We also compared their temporal and spatial neuronal activity properties with those in the extrastriate visual cortex to infer how self-motion signals may be reserved or varied when they are propagated from parietal-temporal areas to the hippocampal system via the hub of PCC and RSC.

## Results

Self-motion cues including vestibular or optic flow were provided in either the vestibular-only or visual-only condition respectively through a laboratory-built up virtual reality system (*Figure 1A*). The two stimuli conditions were interleaved across trials in each experimental session. In the vestibular-only condition, to activate the vestibular channel, each trial contains a Gaussian velocity profile with a corresponding biphasic acceleration profile, lasting 1.5 s (*Figure 1B* shows the motion profile in translation and rotation condition, respectively). Animals were translated or rotated through the motion platform along 26 directions or axes that were equally distributed in three-dimentional space (*Figure 1C*, see Materials and methods). No optic flow was provided in this condition. In the visual-only condition, the motion platform was stationary, whereas self-motion stimuli were simulated by optic flow, with its motion profile simply following that in the vestibular condition to simulate what is experienced on retina during physical motion in the environment. In the following, we first analyzed and presented data under the vestibular condition, then we compared these data to that in the visual condition in the last section of the Results.

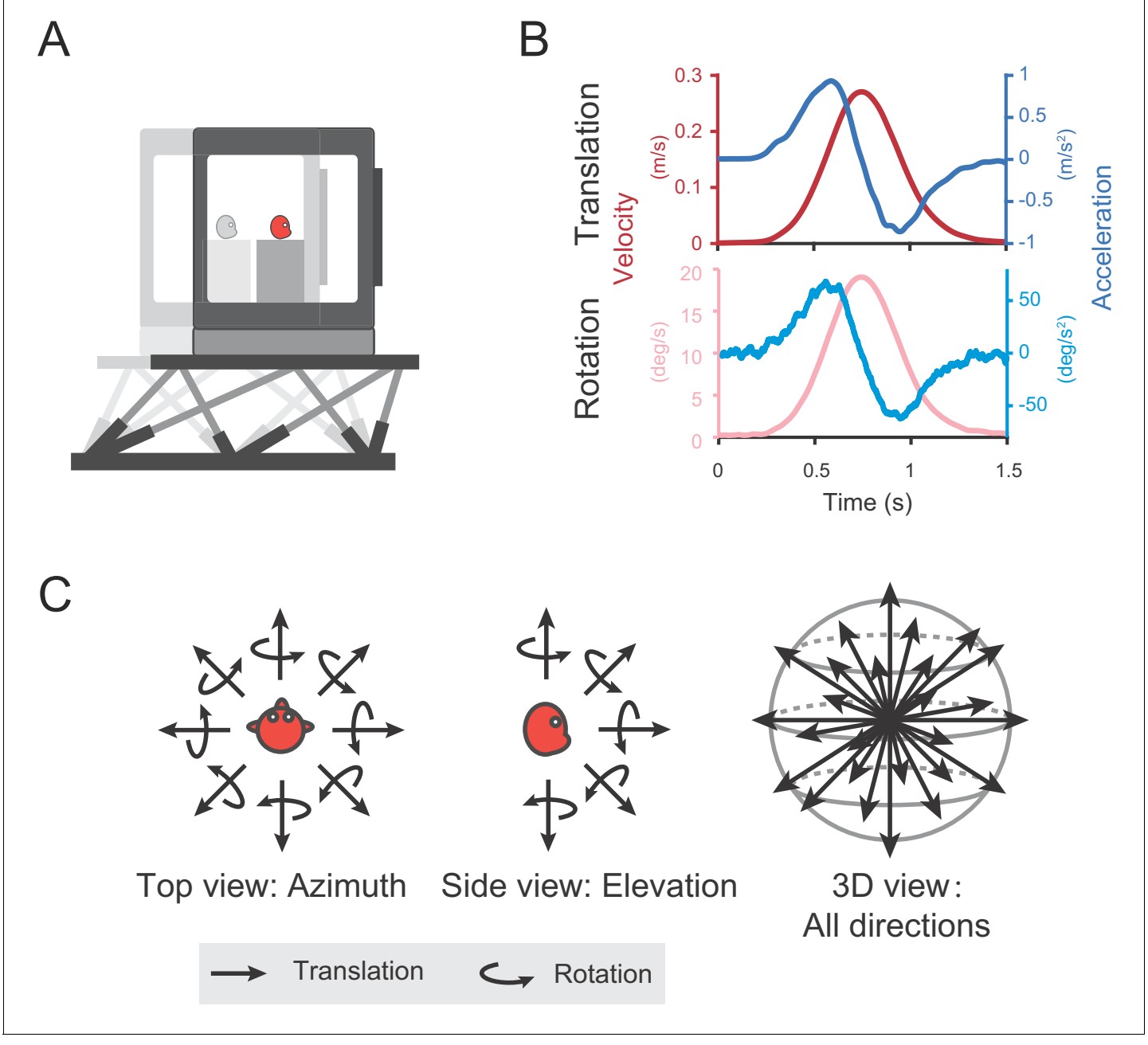

**Figure 1.** Diagram of experimental set-up. (**A**) Monkeys were translated or rotated by a 6-degree-of-freedom motion platform (MOOG). A visual display is mounted on the platform, providing visual stimuli that simulate real motion. (**B**) Motion profiles. For translation, acceleration (dark blue) signals were collected from an accelerometer mounted on the platform and velocity quantity (dark red) is an integral from the acceleration. For rotation, velocity (light red) signals were collected from a gyroscope mounted on the platform and acceleration (light blue) is the derivative from the velocity. Stimulus duration is 1.5 s. (**C**) Illustration of 26 motion vectors in 3D space under translation (straight arrow) and rotation (curved arrow) conditions. Left panel indicates the eight equally distributed directions in the horizontal plane from the top view. Middle panel shows five equally distributed directions in the vertical plane from the side view.

A total of 499 single neurons were recorded by single tungsten microelectrode in the posterior part of the cingulate sulcus above the corpus callosum, including the posterior cingulate cortex (PCC) and retrosplenial cortex (RSC) from five hemispheres of three macaques. Among these neurons, 381 units were identified in PCC (Monkey Q: n = 274, monkey P: n = 89, monkey W: n = 18), and 118 units were in RSC (Monkey Q: n = 84, monkey W: n = 34). Location of recording sites was

identified and confirmed based on combination of MRI scans with simultaneous electrode penetration, atlas, and gray/white matter patterns (*Figure 2*).

## General temporal and spatial modulation properties of vestibular signals

Many neurons in the posterior cingulate region particularly in PCC carried robust vestibular signals yet with heterogeneous temporal and spatial modulation. *Figure 3* showed peristimulus time histograms (PSTHs) from two example PCC neurons under the translation condition. For the first example neuron (*Figure 3A*, upper panels), there were clear activities in response to rightward and upward motion stimuli under the translation condition, with response envelope close to the Gaussian velocity profile, indicating a strong velocity signal. In addition, directions with maximal firing were unvaried across stimulus duration (*Figure 3A*, lower panels). By contrast, the second example neuron as shown in *Figure 3B* exhibited strong peak-to-trough modulation under the translation condition, resembling the biphasic acceleration profile. This type of neurons contained strong acceleration-component signals. Unlike the first example neuron, the maximal firing directions of the second neuron varies across stimulus duration (*Figure 3B*, lower panels). The two example neurons also exhibited robust activities under the rotation contion (*Figure 3—figure supplement 1*). In RSC, we also found a population of neurons carried significant vestibular modulation, albeit less reliable compared to those in PCC. In *Figure 3—figure supplement 2*, the example RSC neurons showed some temporal modulations under both translation and rotation conditions, but the spatial tunings were not truly clear.

To quantify the strength of vestibular temporal and spatial modulations in each area, we conducted our analyses based on the averaged PSTHs across trials in each translation direction or rotation axis. We first used a criterion of temporal tuning (*Chen et al., 2010*) to assess proportion of neurons significantly modulated by the vestibular stimuli in each area (see more detail in Materials and methods). According to this criterion, in PCC, 68% (252/372) cells were significantly modulated by inertial motion under translation condition, whereas 59% (161/271) cells were significantly modulated under the rotation condition (*Figure 4A*, pie graph and *Table 1*). About half of the neurons

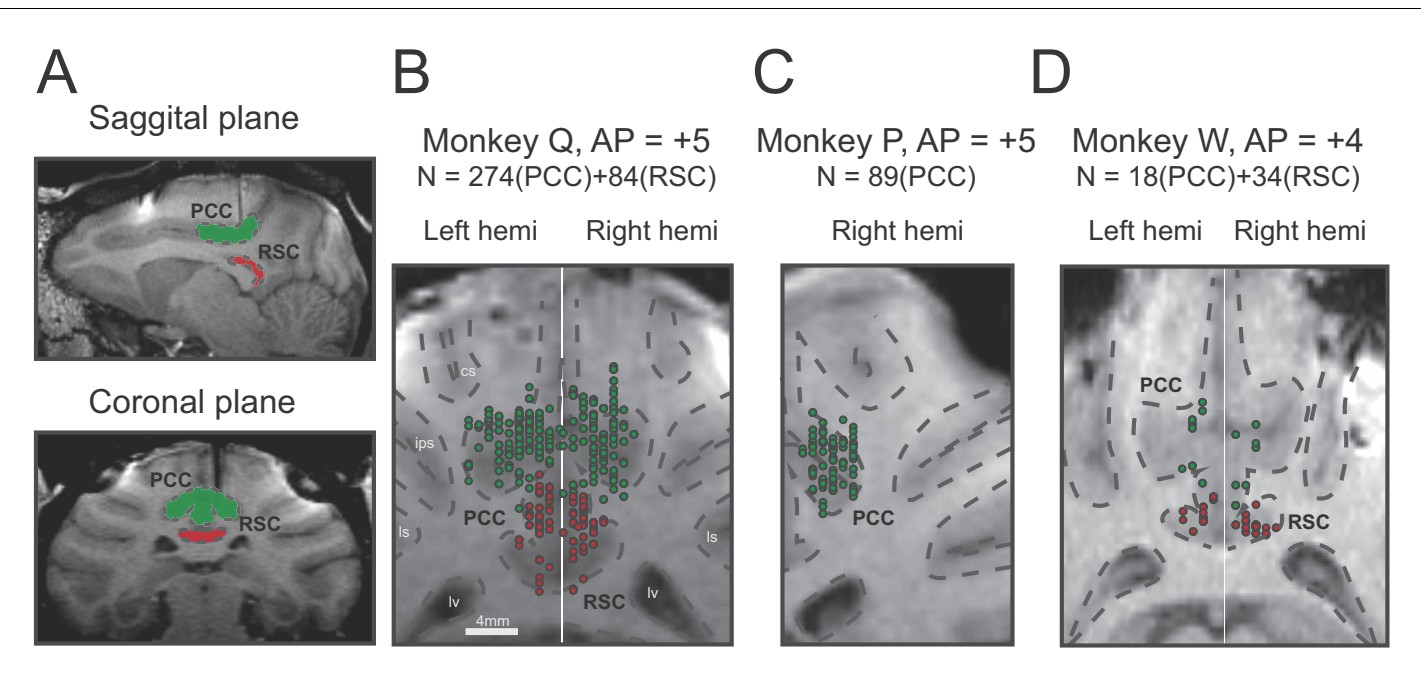

**Figure 2.** Recording sites reconstructed on MRI images of three monkeys. (**A**) Sagittal and coronal planes in one animal (monkey Q), with green shaded areas indicating PCC and red shaded areas indicating RSC. Electrode penetration is indicated by the dark shaded lines. (**B–C**). Recording sites are reconstructed and superimposed onto one plane in each of the three animals. RSC, retrosplenial cortex; PCC, posterior cingulate cortex; cs, central sulcus; ips, intraparietal sulcus; ls, lateral sulcus; lv, lateral ventricle.

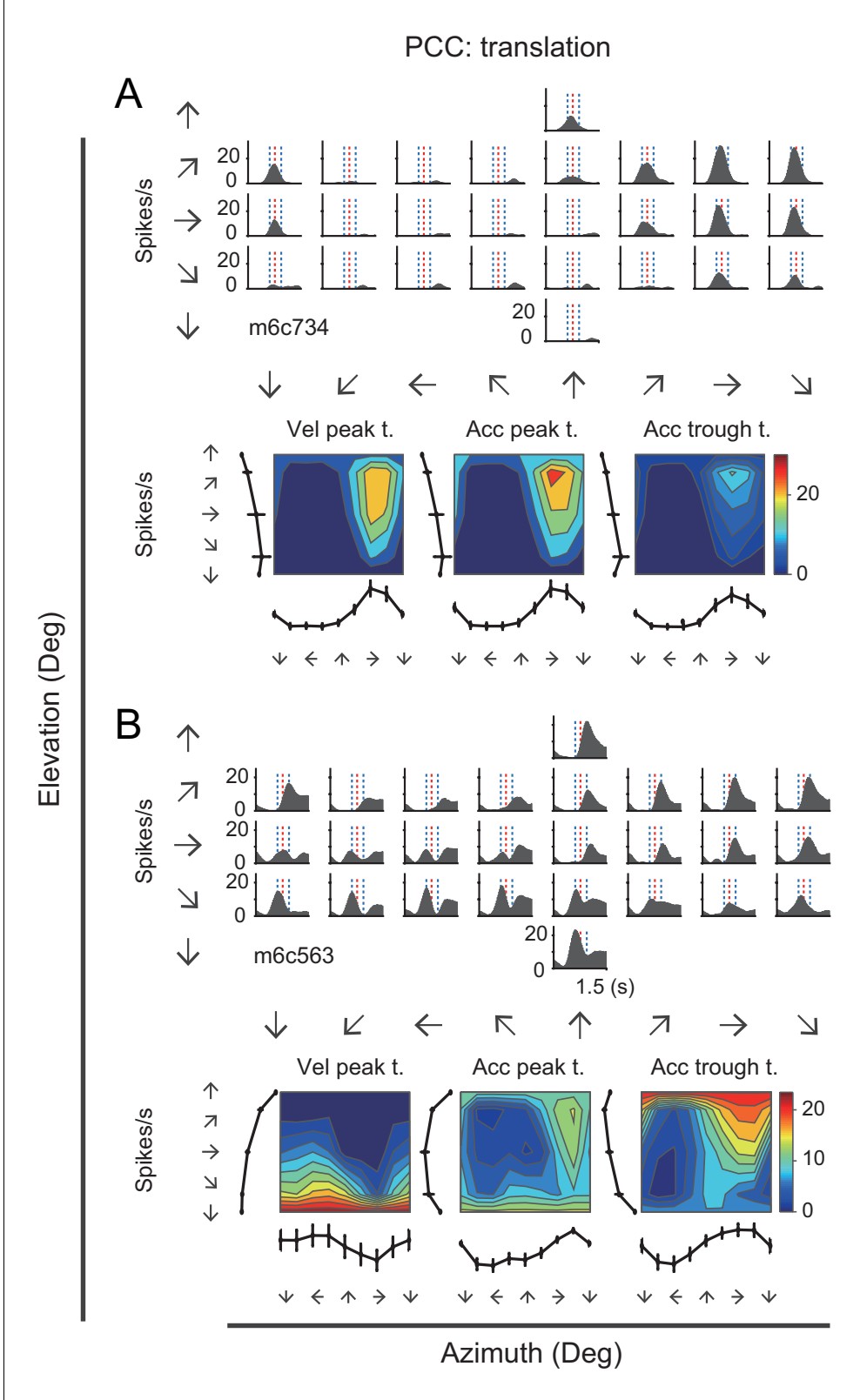

**Figure 3.** PSTHs from two example neurons in the vestibular condition. (**A**) Upper panels: PSTHs of an example PCC velocity-dominant neuron across 26 directions under the translation condition. The red dashed lines indicate the velocity peak time of velocity profile and the two blue dashed lines indicate the peak time of acceleration peak and trough. Bottom panels: Contour figures show firing rates as a function of azimuth and elevation at

*Figure 3 continued on next page*

*Figure 3 continued*

three different time points (velocity peak time, acceleration peak and trough, respectively). (B) Upper panels: PSTHs of an example acceleration-d neuron under the translation condition. Bottom panels: Contour figures of this neuron.

The online version of this article includes the following figure supplement(s) for figure 3:

**Figure supplement 1.** PSTHs from the same two example neurons under rotation condition.
**Figure supplement 2.** PSTHs from two example neurons in RSC.

(50%, 131/263) were significantly modulated by both translation and rotation stimuli, indicating that quite a large proportion of PCC neurons received inputs from otolith and semicircular canals at the same time. Compared to PCC, fewer RSC neurons were significantly modulated under translation (32%, 37/114) and rotation (29%, 25/87) conditions (*Figure 4A*, pie graph). Only 17% (14/83) RSC neurons responded to both translation and rotation stimuli. Thus, more PCC neurons (~2/3) than RSC neurons (~1/3) are temporally modulated by inertial motion stimuli.

We considered these neurons with significant temporal modulations as vestibular tuned cells, and further assessed their spatial tuning across all 26 motion vectors by statistics of one-way ANOVA (see Materials and methods). We found in PCC, a large proportion of neurons showed significant direction tuning ($p < 0.01$, one-way ANOVA) under translational condition (75%, 189/252) and under rotation condition (77%, 124/161, *Figure 4A*, bar graph). Around 71% (93/131) neurons were spatially tuned under both translation and rotation conditions. However, in RSC, only 14% (5/37) and 36% (9/25) neurons were spatially tuned to translation and rotation stimuli, respectively (*Figure 4A*, bar graph). And only 21% (3/14) neurons had significant spatial tuning in both translation and rotation conditions.

After identifying neurons with significantly spatial-tuned responses, we further used a direction discrimination index (DDI, see Materials and methods) to assess the strength of their modulations (*Figure 4B*). DDI ranges between 0 and 1, with values close to one indicating strong spatial preference. The average DDI of all PCC neurons was $0.57 \pm 0.007$ (mean ± S.E.M.) under the translation condition, and $0.58 \pm 0.009$ under the rotation condition. For RSC neurons, the average DDI was $0.49 \pm 0.013$ and $0.50 \pm 0.017$ under the translation and rotation condition, respectively, which was significantly weaker compared to that in PCC ($p = 0.00017$ and $p = 0.0011$ for translation and rotation, respectively, t-test). Thus, neurons in RSC carry much weaker vestibular spatial tuning than that in PCC.

For neurons with significant spatial tuning, we also investigated their preferred directions in 3D space (See Materials and methods). Briefly, spatial preference of each unit was calculated from vector sum of responses in all directions, with azimuth and elevation as spherical coordinates. Under translation condition (*Figure 4C*, dark blue circles), the preferred directions of PCC neurons were distributed almost everywhere in the 3D space, yet there was a tendency that more neurons preferred leftward and rightward motion in the horizontal plane ($p = 0.01$, Uniform test, see Materials and methods). Such a pattern suggests that PCC neurons may be suitable for fine heading discrimination around straight forward during spatial navigation (*Gu et al., 2010*). Interestingly under rotation condition (*Figure 4C*, light blue triangles), the preferred rotary axis also concentrated around the horizontal plane ($p = 0.0225$, Uniform test), corresponding to pitch (i.e. nose-down or nose-up tilt). The functional implications of such a preference were further discussed in the section of Discussion. Finally, for RSC, because there were not enough faction of neurons exhibiting significant spatial tuning, it was not possible to obtain true distribution of their direction preference in the 3D space.

Finally, we examined data and did not find any significant difference among individual animals and hemispheres (*Figure 4—figure supplement 1*, *Figure 4—figure supplement 2*).

## Dark and sound control

To further confirm that the observed vestibular responses in the posterior cingulate region indeed originated from otolith and canals instead of from other cues, we conducted two control experiments.

The first control experiment was to exclude the possibility that the vestibular responses arose from retinal slip due to incomplete vestibular ocular reflex (VOR) suppression under condition when there was residual light in the experimental room. The visual display was thus turned off to remove

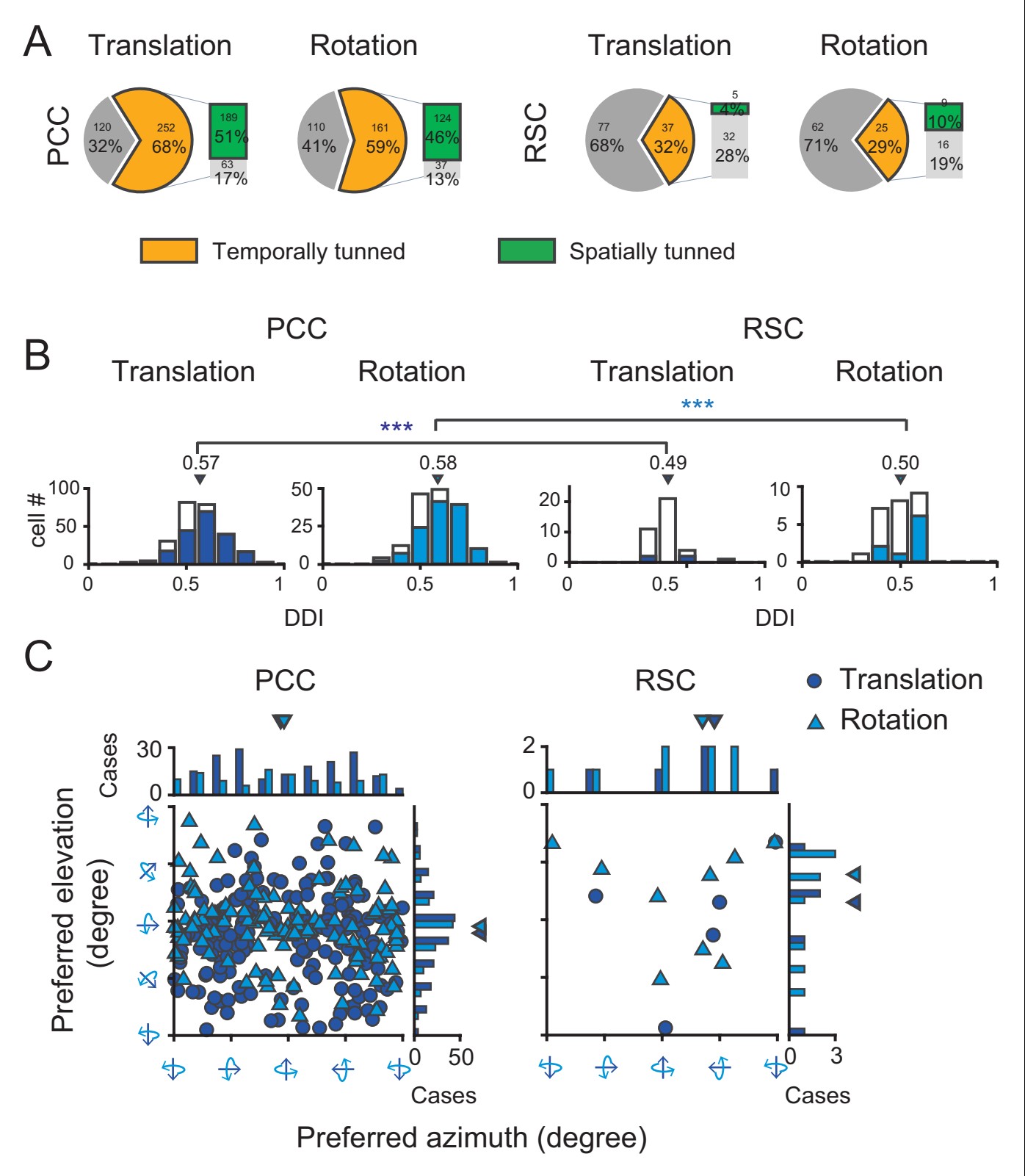

**Figure 4.** Population summary of temporal and spatial tuning properties of vestibular signals in PCC and RSC. (**A**). Proportion of neurons in each category. (**B**) DDI distribution of neurons with significant temporal tuning. Filled bars indicate neurons with significant spatial tuning. (**C**) Distribution of preferred directions for neurons with significant spatial tuning. Dark blue symbol: translation condition; Light blue symbol: rotation condition.

*Figure 4 continued on next page*

*Figure 4 continued*

The online version of this article includes the following figure supplement(s) for figure 4:

**Figure supplement 1.** Population summary of temporal and spatial tuning properties of vestibular signals according to the three monkeys.

**Figure supplement 2.** Population summary of temporal and spatial tuning properties of vestibular signals according to left and right hemispheres.

any possible light in the environment, generating a total darkness condition. Thus, unlike the previous visual-display-on condition in which a central fixation spot was available, the animals were not required to maintain fixation anymore in the total darkness condition. *Figure 5* shows the PSTHs of an example neuron during the total dark condition, as well as during the fixation condition. It is clear that PSTHs in dark condition showed consistent spatial and temporal modulation patterns with those in the fixation condition (r = 0.76, p<<0.001, Pearson's correlation).

We analyzed data based on a population of neurons that were tested under both fixation and darkness conditions. Majority of the neurons showed fairly consistent modulations under the two conditions as indicated by the Pearson's correlation coefficient between the PSTHs in the two conditions (*Figure 6A*, and see *Table 2* for tuning property details). Moreover, DDIs in the two tasks were highly correlated (r = 0.79, p<<0.001, Pearson's correlation), and their mean values were basically the same (p=0.36, paired t-test, *Figure 6B*). Response strength as indicated by the maximal firing rate also had a strong correlation and did not show significant difference between the two tasks (r = 0.80, p<<0.001, Pearson's correlation; p=0.74, paired t-test, *Figure 6C*).

The second control experiment was a sound condition, aiming to exclude the possibility that the observed responses under the vestibular condition was from sound generated from motion platforms during mechanical motion. To do this, we recorded the sounds when the motion platform moved in the vestibular condition, and delivered them from one loudspeaker to the animals when motion platform was stationary in the sound control experiment. As shown by the example neuron in *Figure 5C*, PSTHs in sound condition were dramatically changed, generating a null-modulation pattern that was totally unlike that in the fixation condition (r = −0.07, p=0.0002, Pearson's correlation).

We tested 11 neurons in total. None of these neurons exhibited significant temporal tuning in the sound control experiment (p>0.01, one-way ANOVA, and see *Table 2* for details), and all neurons showed quite weak correlation with the PSTHs in the vestibular condition (*Figure 6D*). Moreover, neither DDI nor Maximal firing rate were close in the fixation and sound tasks (*Figure 6E,F*).

Thus, we exclude the possibility that the vestibular responses observed in the posterior cingulate region are visual responses due to residual retinal slip. We also largely exclude the possibility from an auditory cue, yet a more thorough investigation covering more factors, such as spatial location, binaural effect, loudness of these cues should be conducted in future studies.

## Fitting neural dynamics by a 3D spatiotemporal model

The aforementioned traditional analyses revealed that there were robust vestibular signals in the posterior cingulate sulcus particularly in PCC. These vestibular-tuned cells typically carried complex temporal dynamics that contained both velocity and acceleration components. To further characterize the temporal and spatial tuning properties for different signal components, we fitted responses

**Table 1.** Temporal and spatial tuning properties across subregions of posterior cingulate area under different conditions.

| Subregion | Conditions | Vestibular | | Visual | |
| --- | --- | --- | --- | --- | --- |
| | | Temporally tuned | Spatially tuned | Temporally tuned | Spatially tuned |
| PCC | Translation | 252/372 (68%) | 189/252 (75%) | 114/367 (31%) | 41/114 (36%) |
| | Rotation | 161/271 (59%) | 124/161 (77%) | 84/260 (32%) | 20/84 (24%) |
| RSC | Translation | 37/114 (32%) | 5/37 (14%) | 13/114 (11%) | 0/13 (0%) |
| | Rotation | 25/87 (29%) | 9/25 (36%) | 12/87 (14%) | 1/12 (1%) |

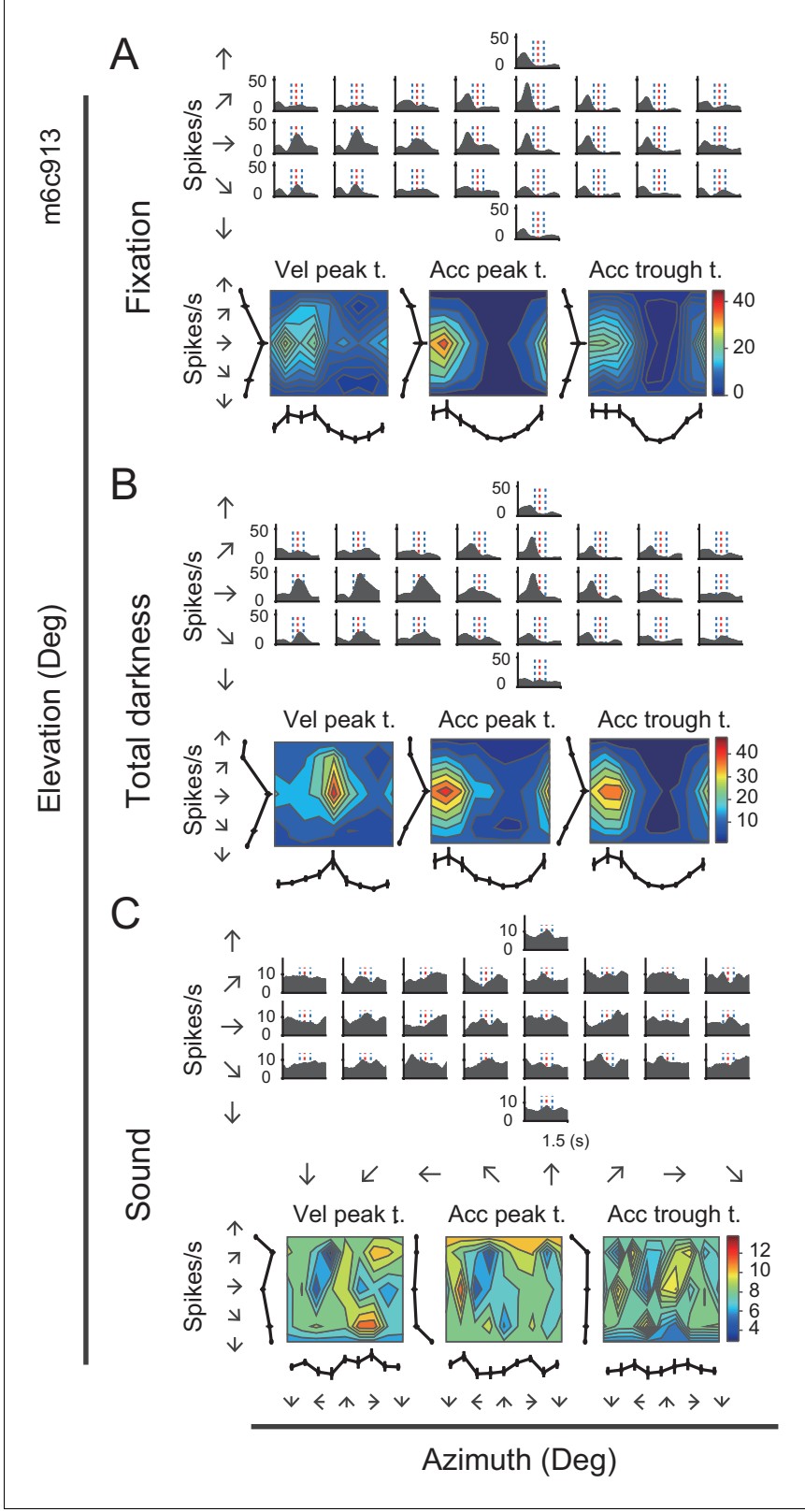

**Figure 5.** Neural activity of an example neuron in total darkness and sound conditions. (A–C) Responses from an example cell in the regular vestibular translation condition (A), total darkness condition (B), and sound control condition (C).

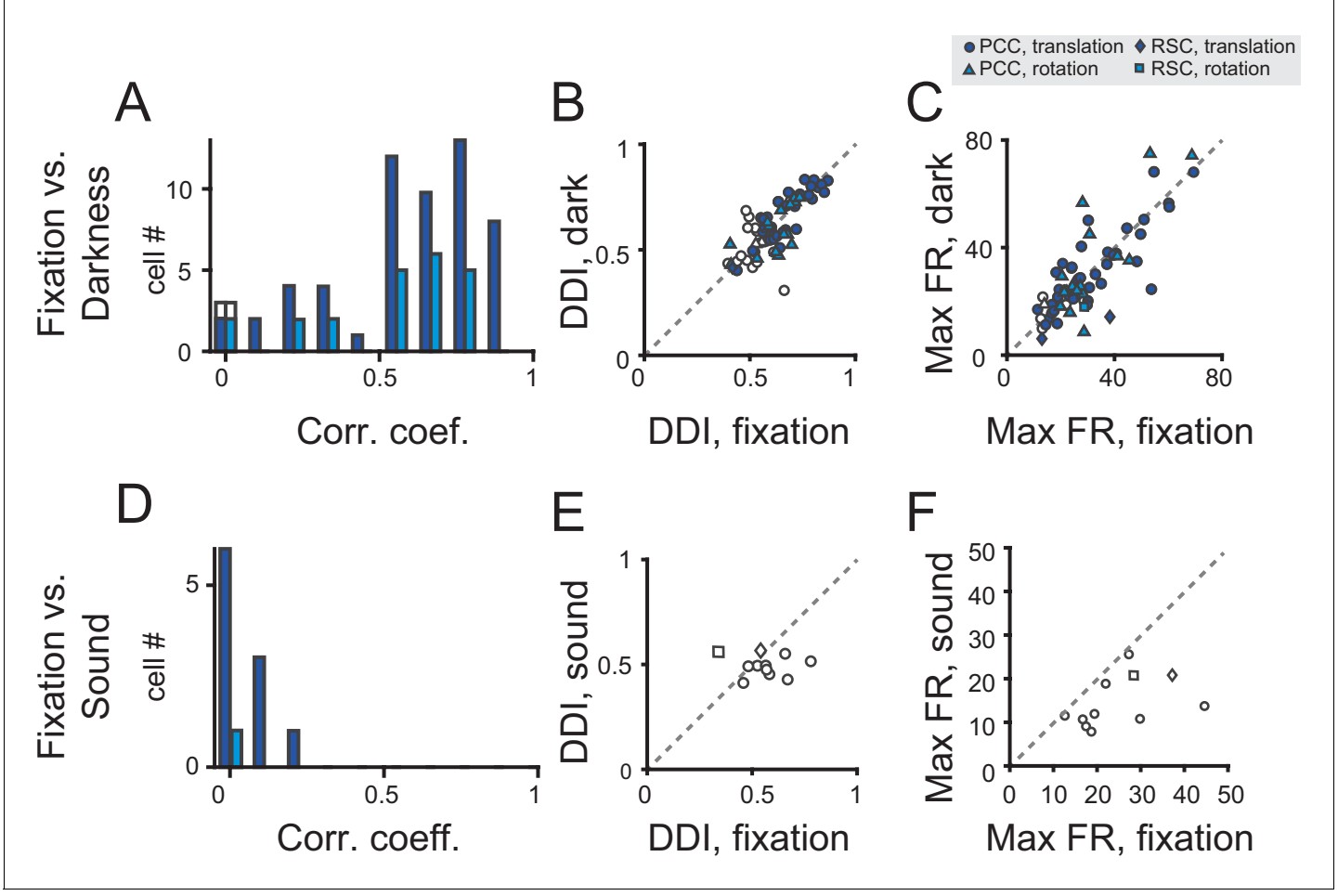

**Figure 6.** Population analysis in total darkness and sound conditions. (A–C) Pearson correlation coefficient of PSTHs (A), DDI (B), maximum firing rate (C) between regular vestibular and total darkness conditions. Filled symbols indicate neurons with significant spatial tuning, whereas open symbols indicate neurons without significant spatial tuning in either stimuli condition. Dark blue symbol: translation condition; Light blue symbol: rotation condition. (D–E) Pearson correlation coefficient of PSTHs (D), DDI (E), maximum firing rate (F) between regular vestibular and sound control condition.

(PSTHs) of each neuron with a 3D spatiotemporal dynamic model. Note that we only considered neurons with significant temporal tuning, thus for PCC, 252 neurons under translational motion and 161 neurons under rotational motion were included. For RSC, only 37 neurons for translation and 25 neurons for rotation were included in this analysis.

The spatiotemporal model was a linear weighted sum of temporal functions with different signal components, with each component multiplied by a spatial function (see Materials and methods and

**Table 2.** Temporal and spatial tuning properties under total darkness and sound control experiments.

Note in the sound control conditions, there is no neuron with significant temporal tuning, so we did not calculate the spatial tuning properties.

| Conditions | Dark control | | Sound control | |
|---|---|---|---|---|
| | Temporally tuned | Spatially tuned | Temporally tuned | Spatially tuned |
| Translation | 42/49 (86%) | 34/43 (79%) | 0/10 (0%) | / |
| Rotation | 16/18 (89%) | 15/16 (94%) | 0/1 (0%) | / |

*Figure 7—figure supplement 1* for details). We first considered a simpler model by including two most salient components of the velocity and acceleration ('VA' model) that were frequently observed in many neurons. A more complex model including more terms would be further illustrated in the next section. Specifically, in the VA model, the velocity temporal kernel was defined by a Gaussian function and the acceleration temporal kernel was the derivative of the velocity function. For the spatial term, it was modeled as a cosine function fed through a linear function that was depended on parameters of preferred azimuth and preferred elevation (see Materials and methods).

*Figure 7* showed PSTHs from two example neurons fitted with VA model. The first neuron (*Figure 7A*) was the same velocity-dominant unit as in *Figure 3A*. This neuron was fitted well by the VA model ($R^2 = 0.88$), as evident by the large overlap between the fitted curve and the responses. Moreover, the fitted weight of the velocity term was much larger than that of the acceleration ($w_V = 0.75$ versus $w_A = 0.25$), indicating a velocity domination signal. As a comparison, the second example neuron (*Figure 7B*) which was the same acceleration-dominant unit as in *Figure 3B*, was also fitted well by the VA model ($R^2 = 0.65$), yet the weight of acceleration component was much larger than that of the velocity ($w_V = 0.24$ versus $w_A = 0.76$), indicating a greater contribution of the acceleration signal.

Across population, we first assessed the goodness of fit by VA model through calculating coefficient of determination between the fitted and measured response ($R^2$). The distributions of $R^2$ in PCC and RSC under translational and rotational conditions were shown in *Figure 8A*. Overall, PCC responses were well fitted by VA model in both conditions, with median $R^2$ value of 0.49 for translation and 0.42 for rotation. The overall response delay ($\tau$) was around 0 ms. However, fitting in RSC was worse ($p \ll 0.001$ in both translation and rotation, t-test), with only four neurons and one showing $R^2$ value greater than 0.5 under translational and rotation condition, respectively. This result was roughly matched with a lack of clear velocity or acceleration signals inspected by eyeballing from the PSTHs of RSC neurons. Thus, in the following analysis, we only included PCC data.

We then characterized contributions of the velocity and acceleration quantity for the posterior cingulate neurons by calculating the log ratio of the velocity weight to the acceleration weight (V/A, *Figure 8B*). To guarantee reliable results, we only selected those neurons with model's goodness of fit ($R^2$) larger than 0.5. In this area, the log V/A ratio distribution tended to be smaller than 0 under the translational condition (median $= -0.21$, $p \ll 0.001$, Wilcoxon rank sum test), while the distribution tended to be larger than 0 under the rotation condition (median $= 0.47$, $p \ll 0.001$, Wilcoxon rank sum test). Thus, PCC neurons in general carry more acceleration signals in translation condition and more velocity signals in rotation condition.

We further analyzed spatial tuning for each temporal component. In particular, preferred direction for each component was defined based on the spatial kernel (*Figure 7—figure supplement 1*). For example, both units as shown in *Figure 7* exhibited similar preferred directions between the velocity and acceleration components (*Figure 7A&B*, lower panels). However, this was not the case across population, similar to that found previously in other brain regions (*Laurens et al., 2017*).

## Jerk and position signals

In addition to velocity and acceleration components, we also noticed some other temporal signals that were missed in the VA model. In one of these, the jerk signal is the derivative of acceleration and defines smoothness of the accelerated motion of the head or whole body in the environment. Previous psychophysical and neurophysiological studies have indicated that such a signal was present in the vestibular afferents (*Fernández and Goldberg, 1976*) and used by subjects to help detect strength of self-motion (*Benson et al., 1986*; *Grant and Haycock, 2008*). The other is the position signal, resulted from the integral of the velocity signal and defines the travelled distance, or rotated head direction, which is critical for path integration (*Etienne and Jeffery, 2004*). Thus, we further fitted neural responses with a more complicated 3D spatiotemporal model including two additional terms of jerk and position based on the previous VA model. The new full model contained four signal components in total (i.e. position, velocity, acceleration, jerk, so-called 'PVAJ' model), with each component having its own temporal and spatial kernel (*Figure 7—figure supplement 1*). For example, *Figure 9* showed PSTHs from two example neurons fitted by the full PVAJ model. The fitted weight of the first unit (*Figure 9A*, upper panels) was $w_V = 0.24$, $w_A = 0.22$, $w_J = 0.42$ and $w_P = 0.13$, among which the weight of jerk was largest. The second unit (*Figure 9B*, upper panels) instead presented a larger weight of position signal: $w_V = 0.08$, $w_A = 0.32$, $w_J = 0.17$ and $w_P = 0.43$.

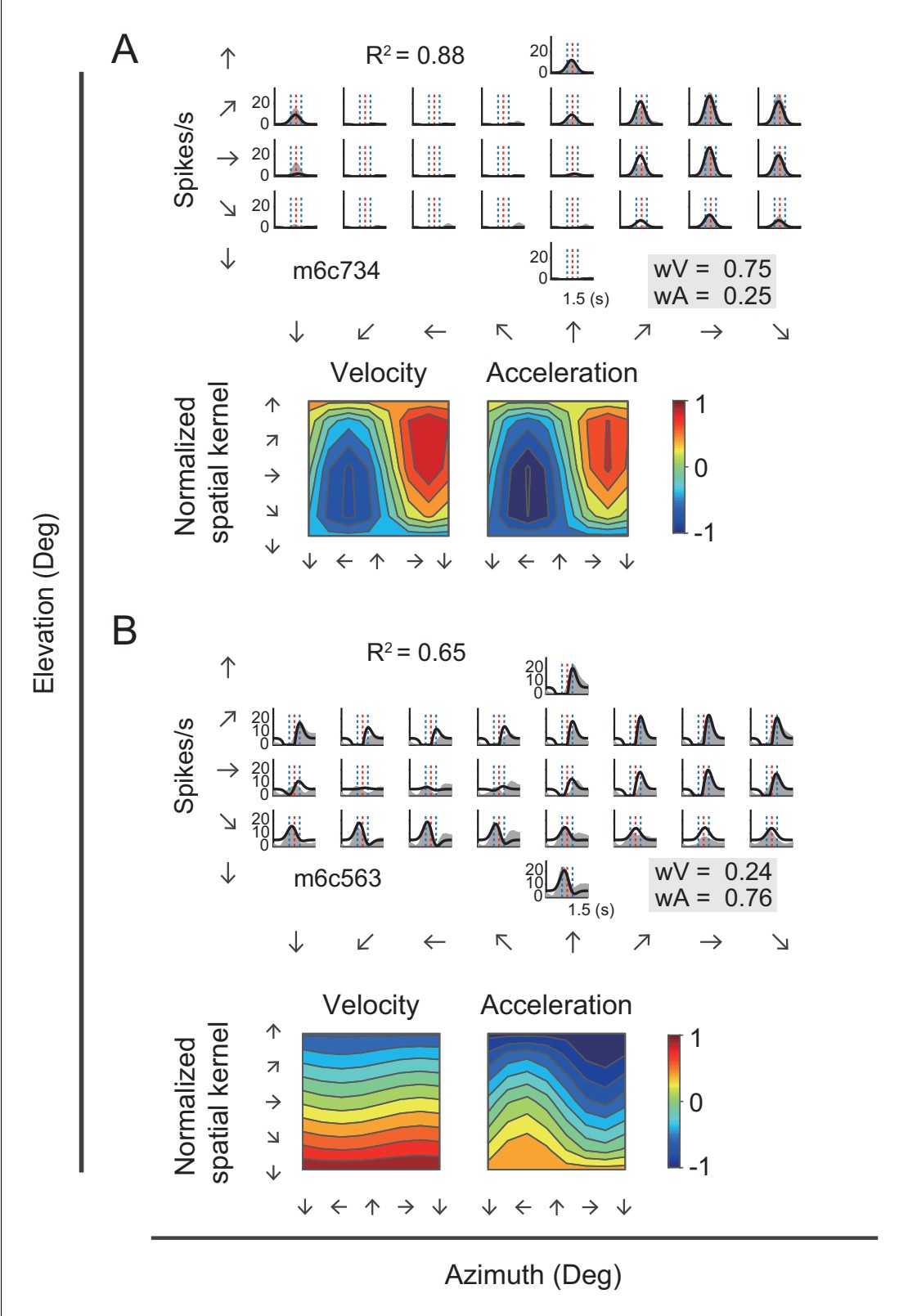

**Figure 7.** Fitting results of two example neurons with VA model. (**A–B**) Upper panels: PSTHs of two example cells fitted by VA model. Gray areas are PSTHs and superimposed black lines are fitted data. Red dashed lines: peak time of velocity; Blue dashed lines: peak and trough time of acceleration. The fitted weight of velocity and acceleration are shown in the gray box. These two examples are the same neurons shown in *Figure 3*. Lower panels: Contour figures of spatial kernels of the velocity and acceleration components.

*Figure 7 continued on next page*

**Figure supplement 1.** Illustration of the three-dimensional temporal-spatial model.

These two example units presented better goodness of fit ($R^2$) of the full PVAJ model compared to the fit of the VA model ($R^2$ = 0.43 and 0.56 versus $R^2$ = 0.69 and 0.75, respectively). This was also the case across the population (p<<0.001 for both translation and rotation in PCC and RSC, paired t-test, *Figure 8A* versus *Figure 10A*). Similar to the VA model, the overall response delay ($\tau$) in the full model was also around 0 ms. However, fitting RSC data with the PVAJ model was much worse compared to fitting in PCC (p<<0.001, t-test, *Figure 10A*), suggesting that vestibular signals in RSC neurons may not simply follow dynamics defined by the classic temporal components, or may not be as reliable as that in PCC. Thus in the following, we will focus on PCC data.

To confirm that the two additional terms of jerk and position quantity really contribute to the neuronal response, we performed two analyses. In the first way, we calculated partial correlation

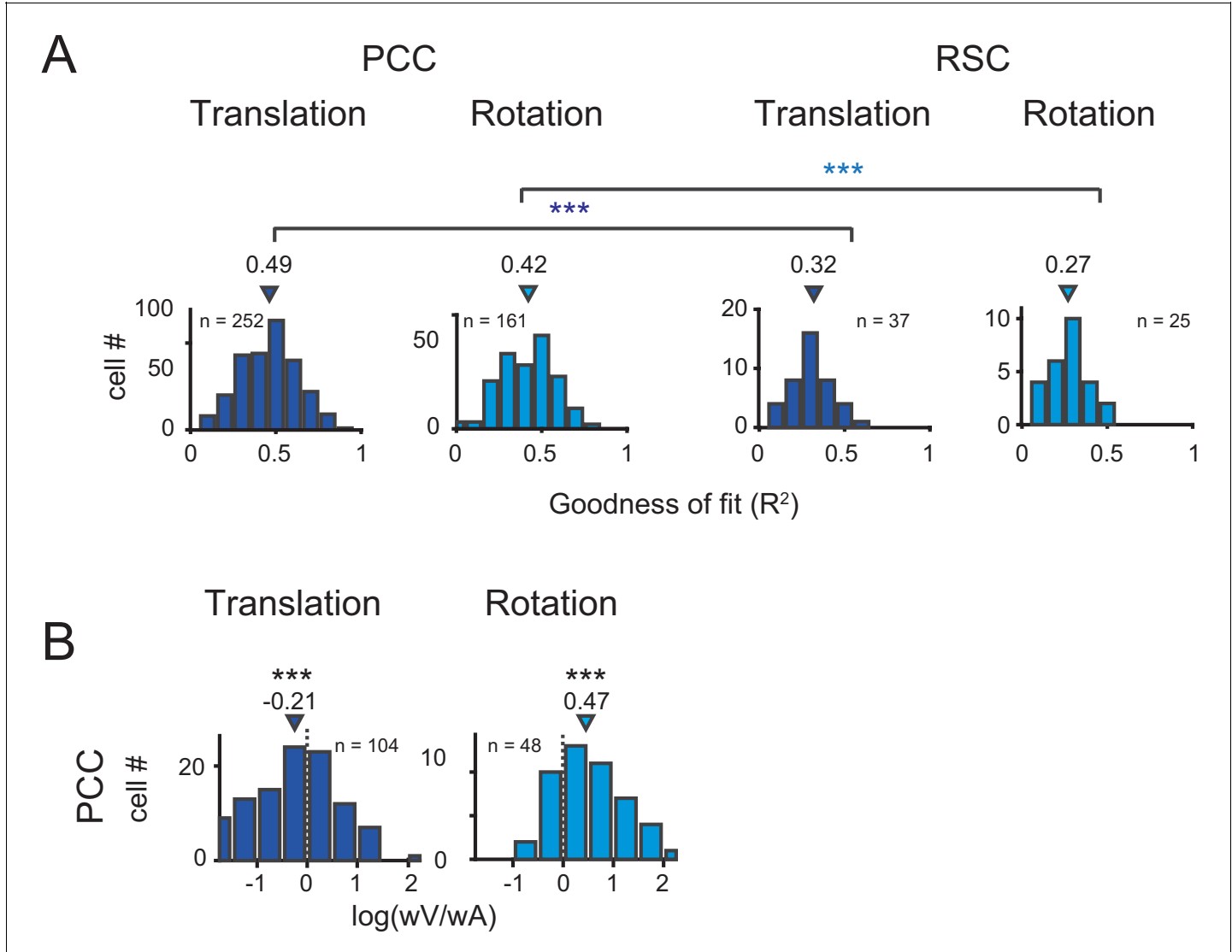

**Figure 8.** Population results with VA model. (**A**) Goodness of fit for PCC and RSC. Triangles are median value of each distribution. (**B**) Log ratio of the velocity weight to the acceleration weight. Triangles show the median value. Only data with $R^2$ >0.5 were included.

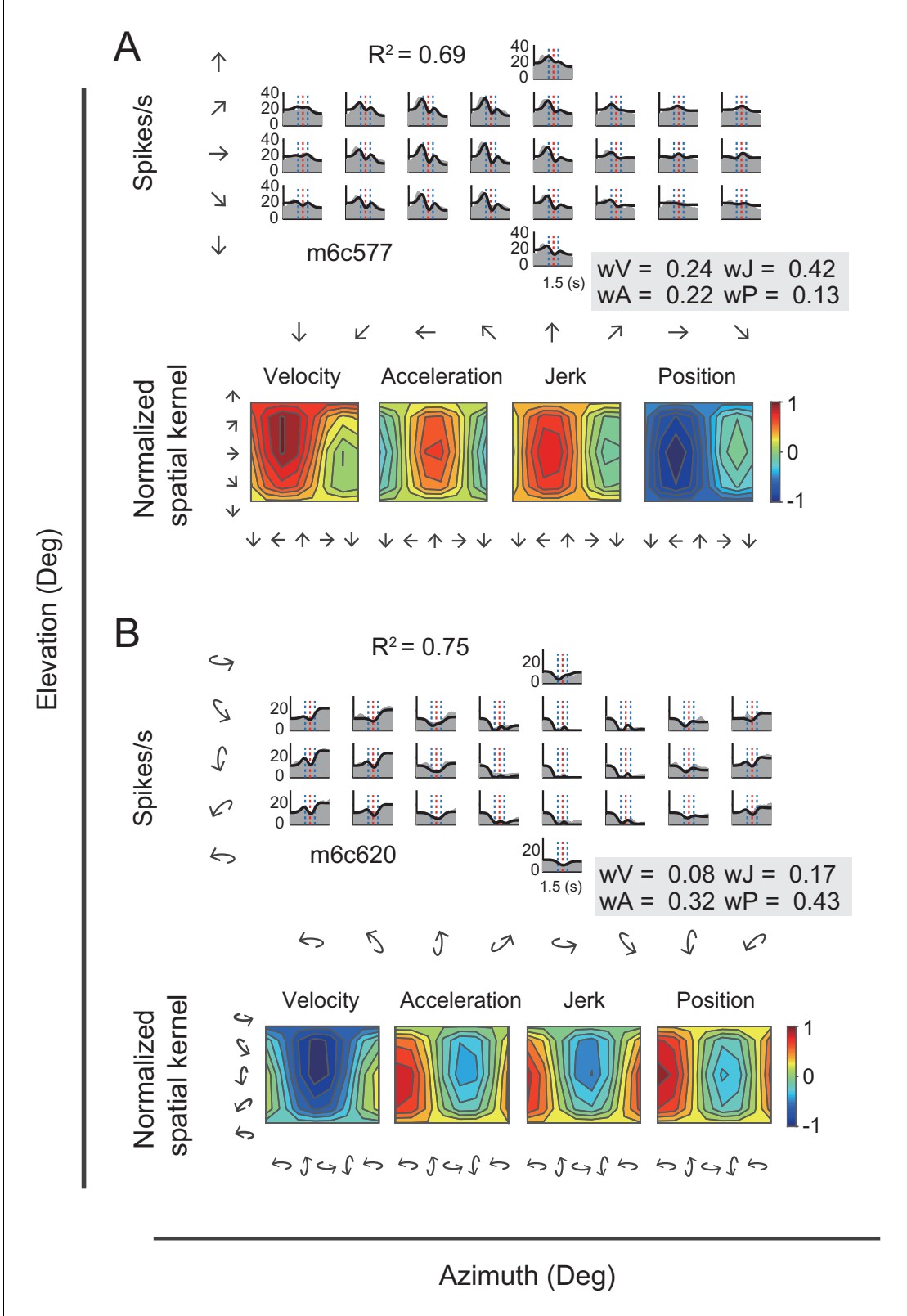

**Figure 9.** Fitting results of two example neurons with PVAJ model. (**A–B**) Upper panels: PSTHs of two example cells fitted by VA model. Gray areas are PSTHs and superimposed black lines are fitted data. Red dashed lines: peak time of velocity; Blue dashed lines: peak and trough time of acceleration. The fitted weight of velocity, acceleration, jerk and position components are shown in the gray box. Lower panels: Contour figures of spatial kernels of the velocity and acceleration components.

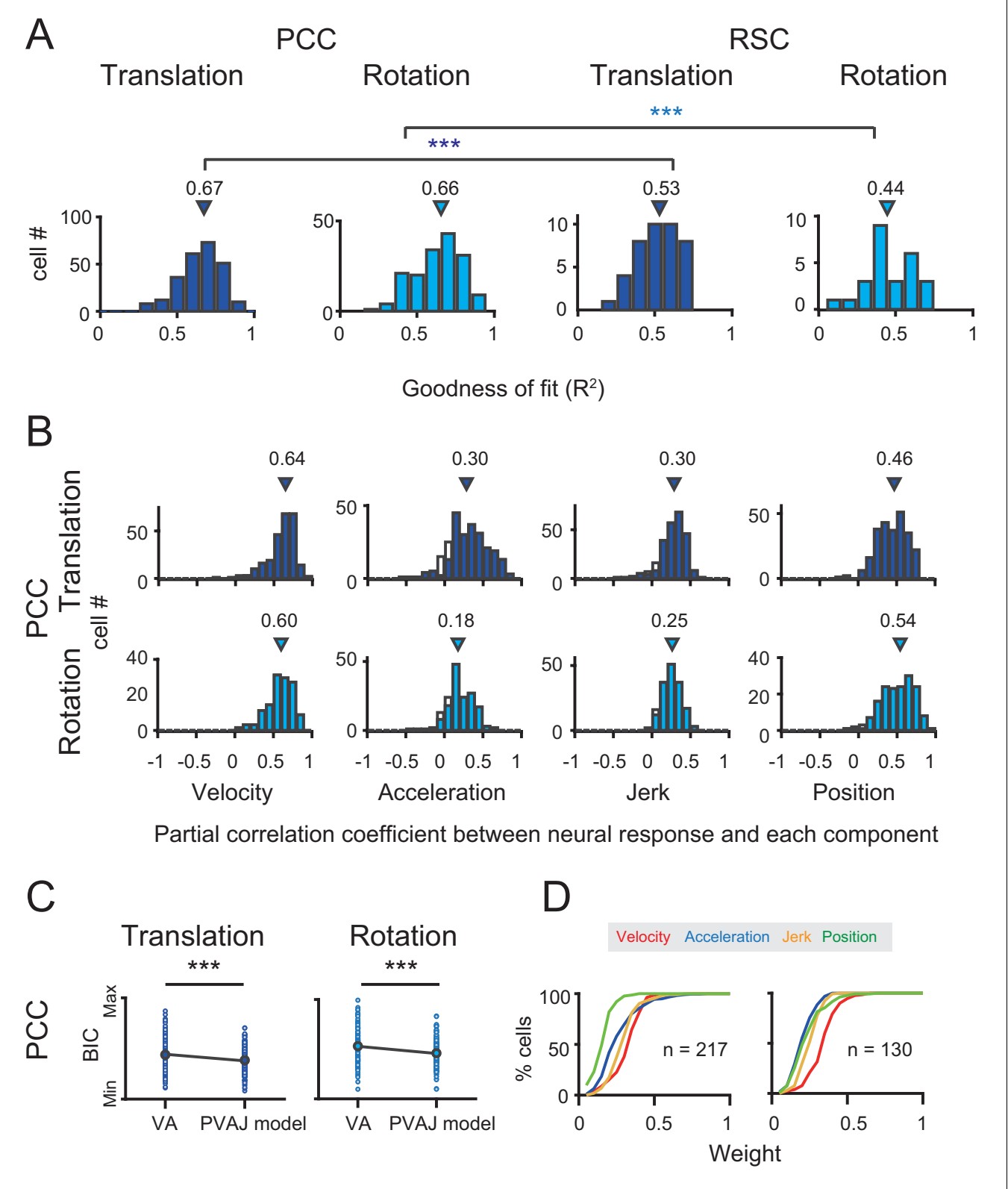

**Figure 10.** Population results with PVAJ model. (**A**) Goodness of fit ($R^2$) from the PVAJ model assessed by Pearson correlation between the experimental and fitted data. Triangles: median value. (**B**) Distribution of partial correlation coefficient between neuronal response and the fitted data of each temporal component in the full model. Filled bar indicates significance with $p<0.05$. (**C**) Comparison of BIC between VA model and PVAJ

*Figure 10 continued on next page*

*Figure 10 continued*

model. The large blue circles indicate mean value that are superimposed on individual values (small circles). (D) Cumulative distributions of the weight of different signal components. Only data with $R^2 > 0.5$ were included.

coefficient between neuronal responses (PSTHs) and the fitted data of each temporal component. As shown in *Figure 10B*, majority of neurons exhibited significant and positive partial correlations for all temporal components ($p < 0.05$, Partial correlation). The average partial coefficients were varied, but comparable across different temporal components, suggesting that the full model captured more variance in the data. In the second way, we addressed the issue from the perspective that any models containing more free parameters would theoretically lead to better fit compared to models with fewer parameters. So we applied Bayesian Information Criterion (BIC) analysis to calibrate this parameter-size effect. As shown in *Figure 10C*, after removing the effect of the number of free parameters, BIC value of PVAJ model was still significantly smaller than that of VA model, confirming that including the jerk and position terms did capture more signals embedded in PCC neurons ($p \ll 0.001$, Paired t-test).

After confirming significant contributions from all four temporal components, we further investigated the relative contribution of each signal captured by the full model. Specifically, proportion of neurons was plotted as a function of each weight value only for those neurons with better goodness of fit ($R^2 > 0.5$, *Figure 10D*). This criterion removed a small proportion of units with unreliable model fits and still reserved a large dataset (translation: n = 217; rotation: n = 130) for the analysis. In particular, the velocity was still more dominant in the rotation condition compared to the other signals which was consistent with that in the VA model (red curves in *Figure 10D*). Acceleration and jerk signals weighed more in the translation condition compared to those in the rotation condition (blue and orange curves respectively in *Figure 10D*). The position signal was less dominant compared to the others, particularly in the translation condition (green curves in *Figure 10D*). Similar to that in the VA model, all temporal components did not show consistent relationships among each other, suggesting that different type of signals may be inherited from different sources.

## Comparison with extrastriate visual cortex

Previous studies have discovered robust vestibular self-motion signals in other cortical areas (*Cheng and Gu, 2018*; *Gu, 2018*). One of these areas that has received heavy studies is the dorsal portion of the medial superior temporal sulcus (MSTd) in the dorsal visual pathway (*Duffy and Wurtz, 1997*; *Bremmer et al., 1999*; *Page and Duffy, 2003*; *Gu et al., 2006*). Population analysis has indicated that MSTd neurons contain robust vestibular signals with velocity-dominant component under translation conditions (*Gu et al., 2006*; *Laurens et al., 2017*). To quantitatively compare temporal dynamics in MSTd with that in the posterior cingulate region, here we used the same spatiotemporal model, including both VA and PVAJ models to fit MSTd responses.

Neural data was fitted quite well by the two spatiotemporal models in MSTd (*Figure 11A*). Consistent with previous findings, Velocity signals tended to be more prevalent in MSTd (VA model: $p \ll 0.001$; PVAJ model: $p \ll 0.001$, Wilcoxon rank sum test, *Figure 11C*), a pattern of which was different from that in PCC. Position and jerk signals were also revealed by the PVAJ model in MSTd. To quantify contributions of these two components, we computed the difference in BIC between the two models (BIC difference index, BDI, see Materials and methods) for both areas. BDI is expected to be larger if including additional terms produces better fit. The median value of the BDI distribution was 0.22 in the posterior cingulate region (only PCC was considered here), and was 0.12 in MSTd (*Figure 11B*, left column). Compared between the two areas, BDI analysis revealed that the full model successfully captured more types of signals in PCC than in MSTd ($p \ll 0.001$, t-test, *Figure 11B*, right column). A closer examination indicated that it was mainly the position signal that was more prevalent in PCC compared to that in MSTd ($p \ll 0.001$, t-test, *Figure 11C*, right columns, green curves). Thus, this finding suggests that posterior cingulate neurons reserve self-motion related signals from upstream areas including the extrastriate visual cortex, but the position signal is further developed here, which may be useful for path integration.

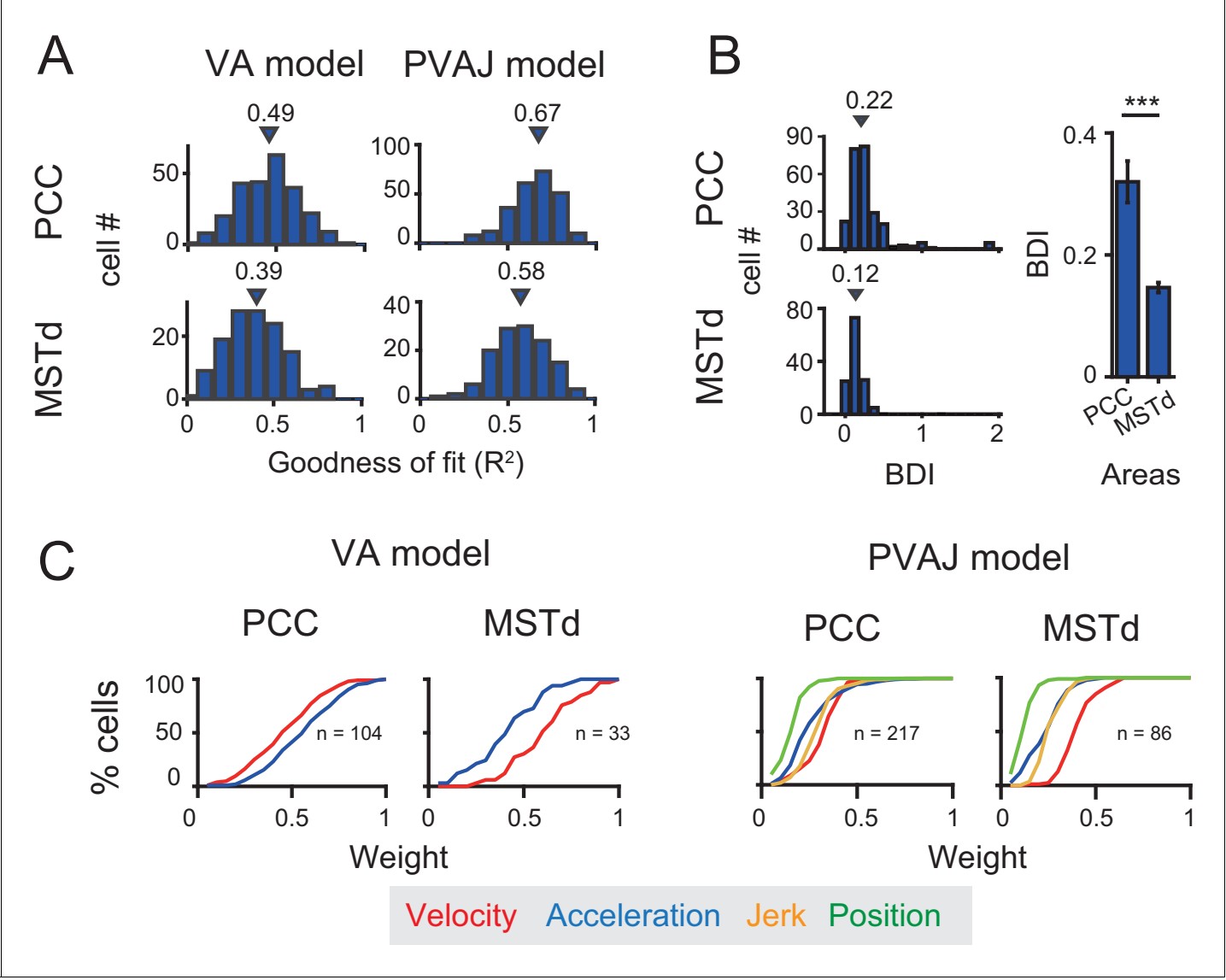

**Figure 11.** Comparison between PCC and MSTd fitted with VA and PVAJ models. (**A**) Distribution of goodness of fit ($R^2$). Triangles: medians. (**B**) Left column: difference in the BIC (Bayesian Information Criterion) between the two models (BDI, BIC difference index, see Materials and methods). Five cases in PCC with BDI larger than two were cut off at two for sake of illustration. Right column: Comparison of the BDI between PCC and MSTd. Bars are the mean value and the error bars are s.e.m. ***: p=0.00034. (**C**) Cumulative distributions of the weight of signal component. Only data with $R^2$ >0.5 were included.

## Visual signals in posterior cingulate region

Self-motion in the environment evokes visual stimuli such as optic flow or array which contains rich information about heading direction or travel distance during spatial navigation (*Gibson, 1950*; *Frenz and Lappe, 2005*; *Frenz et al., 2007*; *Britten, 2008*). In addition to inertial motion, here we also measured neural activities in response to optic flow under the translation and rotation conditions on a large proportion of posterior cingulate cortical neurons (PCC: translation: n = 367; rotation: n = 260; RSC: translation: n = 114; rotation: n = 87). In particular, to simulate real motion, optic flow stimuli containing identical motion profiles as used in the vestibular-only condition were provided to the animals (*Gu et al., 2006*; *Takahashi et al., 2007*). Note that the motion platform was always stationary under this condition so that all the motion information could only be acquired from the visual signals.

Surprisingly, we found that visual self-motion signals were much weaker compared to the vestibular signals in either region (*Figure 12A* and *Table 1*). In PCC, only one third showed significant temporal modulations to optic flow (Translation: 31%, 114/367; Rotation: 32%, 84/260). Furthermore, among those cells with significant temporal modulations, only one third units showed significant (p<0.01, one-way ANOVA) spatial modulations (Translation: 36%, 41/114; Rotation: 24%, 20/84). In RSC, visual responses were even negligible. Only about one tenth neurons exhibited significant temporal modulations (Translation: 11%, 13/114; Rotation: 14%, 12/87), among which almost no units presented significant (p<0.01, one-way ANOVA) spatial modulations (Translation: n = 0; Rotation: n = 1). Thus, across all the cells recorded with an unbiased sampling (see Materials and methods), the proportion of neurons showing significant spatiotemporal modulations in response to visual heading stimuli was fairly small (~10% in PCC, and ~0% in RSC).

We also calculated DDI for the posterior cingulate cortical neurons to assess their visual tuning strength (*Figure 12B*). It turned out that the average DDI was much weaker compared to the vestibular DDI in PCC (p<<0.001, t-test), as well as compared to the visual tuning in the extrastriate visual cortex (e.g. MSTd, *Gu et al., 2006*; *Takahashi et al., 2007*). Hence, visual self-motion signals are overall weak in the posterior cingulate region including both PCC and RSC subregions. Similar to the vestibular signals, visual signals also did not show any significant difference across animals or hemispheres (*Figure 12—figure supplement 1*, *Figure 12—figure supplement 2*).

## Discussion

Numerous studies have indicated that the posterior part of cingulate cortex is involved in many complex behavioral contexts, yet in these studies top-down signals such as attention, anticipation, memory and reward are often mixed with bottom-up sensory driven signals which are difficult to be disentangled from each other. In our current study, by recording spiking activities from these areas in macaques during passive physical motions, we provided solid evidence showing that majority of PCC neurons (~2/3), and a modest proportion (~1/3) of RSC neurons carry robust vestibular linear translation and rotation signals originated from otolith organs and semicircular canals, respectively. A combined 3D spatiotemporal model captured PCC data well and revealed multiple temporal components that could be useful for estimation of instantaneous heading direction or head direction. These properties are consistent with the view that posterior cingulate region may serve as an important hub mediating self-motion related signals propagated from parietal-temporal lobes to hippocampal system for path integration during vector-based spatial navigation (*Rushworth et al., 2006*; *Vincent et al., 2010*; *Kravitz et al., 2011*). Surprisingly though, the same region does not appear to carry robust visual self-motion signals such as optic flow or grating, indicating that posterior cingulate cortex is a vestibular dominant region.

### Comparison with previous studies

Vestibular signals in the posterior region of cingulate cortex have been indicated in previous studies using functional magnetic resonance imaging (fMRI) techniques. For example, *Smith et al., 2012* reported clear vestibular signals in a homologues area (cingulate sulcus visual area, CSv) in the human brain. Other researchers using similar techniques also observed similar activity in similar areas (*Bense et al., 2001*; *Shinder and Taube, 2010*). In these imaging studies, vestibular stimuli were typically provided through caloric or galvanic stimulations due to restraint of head positions during scanning. To achieve more real inertial motion stimuli, *Schindler and Bartels, 2018* adopted a clever method by physically rotating the head of human subjects in the scanner, and measured the blood oxygen level dependent (BOLD) signal immediately after termination of head motion. While these imaging studies provide valuable information of identified vestibular-related signals across the whole brain, there are limitations. In particular, caloric or galvanic stimulation techniques can only activate the vestibular system in a general way (*Suzuki et al., 2001*; *Fitzpatrick and Day, 2004*; *Utz et al., 2010*), instead of being able to simulate real and accurate self-motion including translation along certain directions or rotation around specific axis in 3D environment. Moreover, vestibular signals typically carry complex temporal dynamics (i.e. in milliseconds) that cannot be easily captured by BOLD signals with a relatively slow temporal resolution (i.e. in seconds). Thus, it is important to further conduct studies using single-unit recording techniques to acquire neurophysiological data with much higher spatial and temporal resolutions. In general, we found robust vestibular signals in

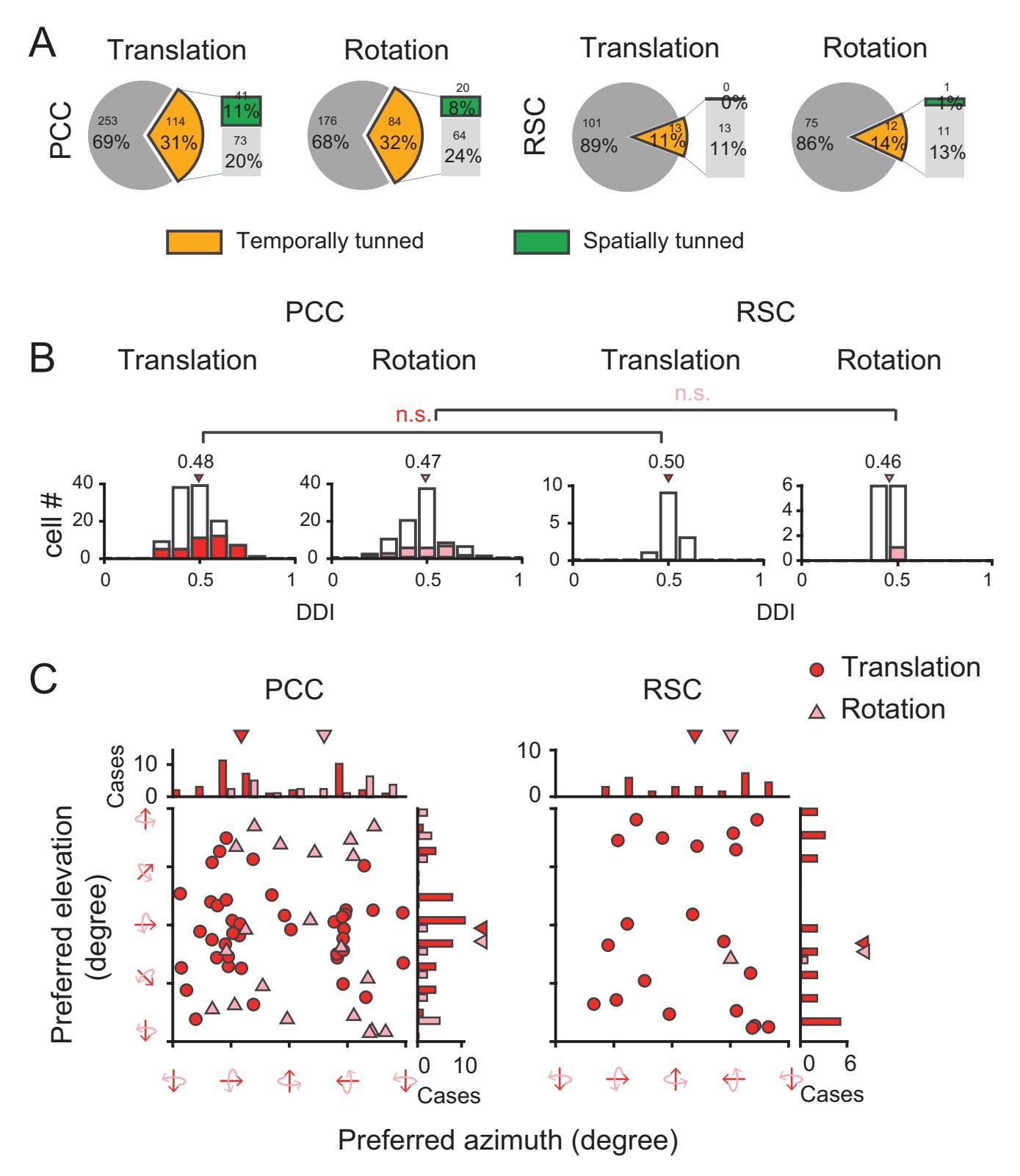

**Figure 12.** Visual self-motion signals (optic flow) in the posterior cingulate region. (**A**) Proportion of temporal and spatial tuning in PCC and RSC. (**B**) Distributions of DDI. (**C**) Preferred direction. Symbol representations are same as in *Figure 4*.

The online version of this article includes the following figure supplement(s) for figure 12:

*Figure 12 continued on next page*

*Figure 12 continued*

**Figure supplement 1.** Population summary of temporal and spatial tuning properties of visual signals according to the three monkeys.

**Figure supplement 2.** Population summary of temporal and spatial tuning properties of visual signals according to left and right hemispheres.

similar regions as indicated in previous imaging studies (yet with more spatiotemporal information as discussed in the following sections), confirming a rough consistency between spiking activity from single neurons with BOLD signals (*Berens et al., 2010*).

However, Smith and colleagues also reported robust visual signals in human cingulate sulcus (*Wall and Smith, 2008*; *Fischer et al., 2012*), as well as in macaque PCC (*Cottereau et al., 2017*). By contrast, we discovered much weaker visual responses using global optic flow or grating. Our results seem to be more consistent with another study, in which BOLD signals were suppressed in response to random motion and static visual stimuli (*Pitzalis et al., 2013*). The inconsistency of visual responses reported across studies may be due to different stimuli used in each experiment. For example, researchers observed visual activity in the posterior region of cingulate cortex when complex optic flow patterns were dynamically changing over time (e.g. switching back and forth between expansion and rotation) (*Wall and Smith, 2008*; *Cottereau et al., 2017*). These activities were basically gone when patterns of optic flow became consistent and stable (*Smith et al., 2017*). It is possible that the posterior cingulate region is more involved in more complex environment when visual stimuli are ever-changing. Other possibilities include specific parameters in the stimuli, for example, the range of temporal frequency of visual stimulus which have been limited in our experiments (see Materials and methods). More types of stimuli and temporal frequencies should be used to test this hypothesis in future experiments.

## Functional implications of vestibular signals in PCC

Direct inputs of vestibular signals to PCC originate from a number of sources, including the thalamus (*Shinder and Taube, 2010*), the parietal insular vestibular cortex (PIVC) (*Guldin and Grüsser, 1998*), and the parietal-temporal cortices (*Pandya et al., 1981*; *Vogt et al., 1987*; *Musil and Olson, 1988*; *Cavada and Goldman-Rakic, 1989*; *Olson and Musil, 1992*; *Akbarian et al., 1994*; *Shinder and Taube, 2010*). Previous studies have demonstrated that vestibular signals with diverse temporal and spatial modulations are ubiquitously distributed across many areas, pointing out a vestibular network in sensory cortices (*Guldin and Grüsser, 1998*; *Laurens et al., 2017*; *Cheng and Gu, 2018*; *Gu, 2018*).

In the current study, we also found clear temporal and spatial modulations in a large proportion of neurons in PCC, with many properties qualitatively similar to those in the insular and parietal-temporal areas, indicating PCC is one of the nodes within the cortical vestibular network. For example, preferred directions of PCC neurons in response to linear translation overall have a bias toward leftward and rightward motion in the horizontal plane, a pattern of which has been seen in several parietal-temporal areas such as MSTd (*Gu et al., 2006*). Information theory predicts that such a pattern of distributed preferred directions is useful for detection of fine-varied headings around straight ahead (*Gu et al., 2010*), suggesting that PCC could be involved in heading perception. As to rotation signals, the distributed preferred axis tends to bias around pitch and roll, but not yaw. A previous study has reported that in mice, azimuth-tuning of head direction cells is affixed to the head-horizontal plane when the animals' head positions were changed relative to gravity (*Angelaki et al., 2020*). Therefore, these rotation signals could be essential for head direction cells to function properly in 3D environments.

We found all temporal components play important roles through partial correlation analysis. Besides velocity, acceleration and jerk which were studied in previous work, our study also captured a position signal in some posterior cingulate neurons. According to weight value, this component is less dominant compared to the others, but is more prevalent than that in the extrastriate visual cortex, suggesting that a process of integrating the velocity quantity to position may happen in PCC when self-motion related information is propagated closer to the hippocampus system.

As to the outputs of PCC, it also directly sends signals back to the vestibular nucleus (VN) in brainstem (*Akbarian et al., 1994*). This pathway may be involved in gaze control or other reflex behaviors during coordinated eye, head, and body movements during natural navigation (*Dean and*

*Platt, 2006*; *Shinder and Taube, 2010*). Thus, as postulated by previous researchers, PCC '. . . *do not serve a simple or direct role either in sensory information processing or in motor control. . .*' (*Vogt and Gabriel, 1993*). Instead, vestibular signals with rich temporal and spatial information in PCC may compose a high and mixed dimensional neural representation that could be used by downstream neurons through a simple, linear readout algorithm to compute an efficient and flexible variable according to ongoing task's need.

Among the brain limbic system, PCC is the one that has been less studied and its exact functions to date remain unclear and debatable compared to the other parts, for example, its counterpart at the anterior cingulate cortex (ACC), an area of which is identified to be involved in emotion processing (*Vogt and Gabriel, 1993*). Both anterior and posterior parts of cingulate cortex are heavily connected with each other. PCC is also reciprocally connected with PIVC that in turn receives projections from the viscera system through the solitary tract and the parabrachial complex (*Vogt, 2019*). Electrical stimulation in PIVC evokes unpleasant emotions including vomiting, nausea, dizziness, and pain (*Penfield and Faulk, 1955*; *Penfield, 1957*; *Mazzola et al., 2012*). Thus, vestibular signals in PCC may also be related to emotion, which requires further investigations in the future. In addition, PCC has been indicated to play roles in many other cognitive functions including default mode network (DMN), visual spatial map, action, learning, spatial memory, decisions (see reviews and books by *Vogt et al., 1992*; *Vogt and Gabriel, 1993*; *Vogt and Laureys, 2005*; *Fransson and Marrelec, 2008*; *Leech and Sharp, 2014*). For example, in a recent study, Gold and his colleagues reported that PCC neurons contain spatial choice and reward-target signals in a complex, adaptive task (*Li et al., 2019*). Future studies need to be conducted for investigation of how vestibular signals in PCC are involved in these functions.

## Self-motion signals in macaque RSC

Similar to PCC, RSC has also been indicated to be involved in spatial navigation tasks albeit mainly in human and rodent studies (*Cooper et al., 2001*; *Maguire, 2001*; *Spiers and Maguire, 2007*; *Vann et al., 2009*; *Clark et al., 2010*; *Cullen and Taube, 2017*). Vestibular yaw-rotation signals have been observed in mice RSC (*Keshavarzi et al., 2021*), as well as primary visual cortex (V1) which may originate directly from RSC as revealed by tracing and functional mapping (*Vélez-Fort et al., 2018*). At the same time, mice RSC also receives heavy inputs from V1 as well as other sensory information related with spatial perception, suggesting RSC is a multisensory area albeit biased toward visual (*Zingg et al., 2014*). Recently, Powell and colleagues also reported visual responses modulated by grating stimuli in mice RSC (*Powell et al., 2020*). By contrast, although we discovered that about one-third RSC neurons in macaques exhibited significant temporal modulations to physical translation or rotation stimuli, visual modulations overall by optic flow appeared to be fairly weak in this area (*Figure 12B*). To verify that we have not missed other visual motion signals, we also introduced the visual gratings stimuli. Similar to optic flow, the responses to grating stimuli is also quite weak in this region (data not shown). A number of factors may lead to heterogeneous results across studies.

First, previous studies on human found strong involvement of RSC in virtual navigation task or when animals were imagining or planning route, which required many high-level top-down signals such as attention, anticipation, or action (see reviews by *Maguire, 2001*; *Spiers and Maguire, 2007*; *Vann et al., 2009*). By contrast, in our study, the monkeys performed a simple fixation task while they were passively experienced self-motion stimuli, a process of which mainly involved sensory-driven signals. Second, previous studies suggest that RSC may be more involved in landmark-based navigation instead of vector-based navigation (i.e. path integration). For example, lesion of RSC significantly impaired mice's performance during spatial navigation when they relied on visual landmarks but not much when animals were required to rely on self-motion cues (*Clark et al., 2010*; *Mitchell et al., 2018*). Thus, RSC may not be much involved in vector-based navigation particularly when relying optic flow to update one's self-motion directions or travelled distance. Third, the exact area of RSC has been defined differently across species. In rodents, RSC (usually defined as RSP) is a relatively large region and PCC is small, whereas in monkeys the region of PCC is very large and RSC is small. In human, both areas are fairly small. Thus, homologous areas across species could be somehow different and execute heterogeneous functions to adapt to environments during evolution.

Compared to PCC, we discovered that vestibular signals overall are less clear in macaque RSC. Close to one-third proportion of RSC neurons, which is about a half of that of PCC neurons, exhibit significant temporal modulations. Moreover, the same 3D spatiotemporal model does not fit data in RSC as well as that in PCC, suggesting that temporal component signals in RSC may either be less reliable, or more complex that need other models to be further developed to better capture the data. Unlike PCC, spatial tuning in RSC neurons is basically lacking. This result may seemingly surprising compared to recent findings in rodents (*Keshavarzi et al., 2021*). However, in addition to the possibilities described above, there is also another one that our motion stimuli contain a smaller rotary speed (maximal ~20°/s) compared to that used in previous rodent studies (maximal ~80°/s). Rotation with high velocity may evoke more responses which should be tested in the future. In any case, our results suggest that macaque PCC and RSC may contribute to spatial navigation in different ways. Future studies could be conducted to investigate spatial reference frames of self-motion signals in these regions, and how these signals are functionally linked to the animals' perceptual decisions during spatial-related tasks.

## Materials and methods

### Subjects and system set-up

We performed the experiments on three male monkeys (*Macaca mulatta*), weighted 8–12 kg (monkey Q, P, and W). The monkeys were chronically implanted with a circular molded lightweight plastic ring above cranium for head fixation and recording, and scleral coils for monitoring eye movements in real time inside a magnetic field with phase detector (Riverbend instrument).

During behavior training and neural recording, monkeys were comfortably seated in a chair, which was secured to a six-degree of freedom (heave, surge, and sway for translation; pitch, roll, and yaw for rotation) motion platform (MOOG, 6DOF2000E) with an attached LCD screen in front of it (*Figure 1A*). With such set-up, we could flexibly switch different stimuli types, vestibular stimuli from the motion of platform and visual stimuli from the screen given by an OpenGL accelerator board, depending on different trial condition. To simulate the visual pattern of real motion projected onto the retina, visual stimuli are depicted as a three-dimensional cuboid space full of $0.3 \times 0.26$ cm yellow triangles (just like stars in the star sky), with 100 cm wide, 150 cm high, 160 cm deep and the density of $0.002$ /cm$^3$. The optic flow field is a simulated 3D space with motion parallax and size information that is dependent on depth. The monkeys were head-fixed and sat at a distance of about 30 cm from the screen, with a visual angle of approximately $90° \times 90°$.

After monkeys were well-trained to the tasks, a recording grid was tightly settled inside the plastic ring horizontally. The grid contains small holes arranged in array with a diameter of 0.5 mm and the holes were apart from each other by 0.8 mm. With the help of the grid and figures from MRI, we can map the position of each site which we performed recording on. All animal procedures were approved by the Animal Care Committee of Shanghai Institutes for Biological Sciences, Chinese Academy of Sciences and have been described previously in detail (*Gu et al., 2006*).

### Experimental protocol

During each task, there was a small fixation point ($0.2° \times 0.2°$) appeared at the center of the screen from the beginning and disappeared till the stimulus was vanished. The monkeys needed to saccade to the fixation point and maintain fixation for 200 ms to initiate the self-motion stimuli. The animals maintained fixation for the following 1.5 s within an electronic window ($3° \times 3°$) before the stimulus was removed. Monkeys were rewarded with juice at the end of trial if they completed the trial successfully, otherwise the trial would stop and the data were aborted.

To characterize basic tuning properties in posterior cingulate region, two sensory modality conditions including vestibular and visual stimuli were conducted in the current study. In the vestibular condition, motion platform was moving along a straight line (i.e. translation condition) or rotating around the center of the interaural axis (i.e. rotation condition) in 26 directions that were equally distributed in 3D space (*Figure 1C*). Only a fixation point was presented on the visual display for the animal to maintain fixation at a head-fixed target (i.e. suppression of vestibular ocular reflex, VOR). In the visual condition, the motion platform was stationary while visual stimuli (optic flow) were

presented on the screen, simulating real motions that were delivered in the vestibular condition. Visual and vestibular stimuli conditions were interleaved in one experimental block.

Motion profiles in either stimulus condition followed a Gaussian velocity, with a corresponding biphasic acceleration profile. Actual velocity and acceleration parameters were acquired from an accelerator and a gyroscope mounted on the motion platform (*Figure 1B*, dark blue line indicates the measured acceleration of translation and light red line indicates the measured velocity of rotation.). By estimation, the peak velocity was ~0.3 m/s and the peak acceleration was 1 m/s$^2$ in the translation condition. In the rotation condition, the peak angular velocity was ~20 °/s and the peak acceleration was 60°/s$^2$.

To make sure that responses under the vestibular condition were not evoked from other cues, we also conducted two additional control experiments. In the total darkness condition, we turned off the visual display to eliminate any residual visual cues in the vestibular condition that might arise from incomplete suppression of VOR. The monkeys were free to make any eye movements in this condition. In the sound control experiment, we tried to test the possibility that the vestibular responses might arise from the sound cue generated from physical motions of the mechanical actuators (peak magnitude: ~90 dBs). To do this, we first put a recorder beside the animal's ears and recorded the sounds when the motion platform was moving with the motion profiles used in the vestibular condition. We then played these sounds to the animals when the motion platform was stationary.

Under the visual condition, for a subpopulation of neurons, we ran an additional block using drifting gratings. Specifically, gratings with a full black-white contrast were moving along eight directions orthogonal to their orientations in the fronto-parallel plane. The temporal frequency is 5 Hz and the spatial frequency is 0.2 and 0.4 cycles/degree in the experiment.

While the animals were experienced self-motion stimuli, single-unit activity was recorded from PCC and RSC in the posterior cingulate region. To validate the position of PCC and RSC, we first used MRI to help identify locations of electrode penetrations. Electrodes were penetrated in the earth-vertical plane. PCC were located roughly around anterior-posterior (−1~+12) and medial-lateral (0 ~ 10). RSC were located roughly around anterior-posterior (+1~+7) and medial-lateral (0 ~ 6). The MSTd data were recorded in a previous study (*Gu et al., 2006*) and were used here as a comparison with the cingulate data.

After entering the targeted areas, single-unit extracellular recording was performed. Tungsten microelectrodes with impedance around 1 MΩ (FHC) were used. Data were recorded and single units were isolated using a multichannel recording system from Alpha Omega (Israel) and CED (Spike2).

## Data analysis

### Characterizing tuning properties

Peristimulus time histograms (PSTHs) in each stimulus condition was constructed and illustrated with a 25 ms bin and was smoothed with a 300 ms Gaussian filter. To quantify whether a neuron was significantly modulated by the stimulus, a two-sided Wilcoxon rank sum test was used to compare responses during the 1.5 s stimulus duration with the spontaneous activity. The spontaneous activity was based on the time window of 100 ms before and 300 ms after stimulus onset. Responses in the 1.5 s stimulus duration was divided into 60 bins, each of which covered 300 ms and stepped at 25 ms. To avoid false positive results, a neuron was considered to have significant temporal tuning only when satisfying two strict criteria at the same time. First, at least five consecutive bins needed to show significant modulation (p<0.01, Wilcoxon rank sum test). Secondly, the first requirement needed to happen in at least two adjacent spatial directions (*Chen et al., 2010*).

To quantify whether neurons were spatially tuned, we first calculated response in each 300 ms-bin and performed ANOVA analysis across 26 vectors (i.e. translation directions or rotation axes). Responses at a certain bin was considered to be significant when p value was smaller than 0.01. To identify the number of response peak across all bins, we performed an analysis as used in previous studies (*Chen et al., 2010*). Briefly, for all those bins with significant spatial tuning (p<0.01, one-way ANOVA), a local peak was temporarily assigned when it was higher than its neighboring bins. Across all these local peaks, a next step was performed to identify which of them were within one group, meaning one real peak, or were separated from each other, meaning multiple peaks (actually only

two-peak cases were considered whereas tri-peak or higher were not considered because those peaks were usually small and unreliable according to experience). In particular for each pair of neighboring local peaks, Pearson correlation coefficient was measured between the two vectors composed of responses in the 26 directions at the local peak bins. Positive correlation led to conclusion that the compared local peaks belonged to the same one peak and negative correlation suggested that they belonged to different peaks. If there was only one real peak in the end, a neuron was defined as 'single-peak' cell, and if there were two peaks, it was defined as 'double-peak' neuron.

To quantify the strength of direction tuning, we used a direction discrimination index (DDI) given by *Takahashi et al., 2007*, as follows:

$$\text{DDI} = \frac{R_{\max} - R_{\min}}{R_{\max} - R_{\min} + 2\sqrt{\frac{SSE}{N-M}}},$$

where $R_{\max}$ and $R_{\min}$ are the maximal and minimal responses respectively, *SSE* is the sum squared error around mean responses, *N* is the total number of observations (trials), and *M* is the number of stimulus directions (here, *M* = 26). Neurons with strong response modulations will have DDI values closer to 1, while neurons with weak response modulations will have DDI values closer to 0. To compute each neuron's DDI, we first found the maximum response direction based on the full 1.5 s stimulus duration. In that direction, we then found the maximum response time bin in one of the 60 bins (300 ms window with a step of 25 ms), and assigned this bin as the $R_{\max}$. Using this time bin, $R_{\min}$ was assigned for the minimum response bin across the other 25 directions. Permutation test was applied to assess significance of DDI.

For neurons with significant spatial tuning, spatial preference of each unit was calculated from vector sum of responses in all directions, with azimuth and elevation as spherical coordinates. For single-peak neuron, preferred direction was calculated around the response peak time. For double-peak neuron, only the first peak time was used since responses at the first peak were generally larger than the second peak in the opposite directions (*Chen et al., 2010*).

## The 3D spatiotemporal model

A combined spatiotemporal model was constructed to fit PSTH data in 26 directions at the same time. The simplest form of this model is a linear combination of different temporal components. Each term is a multiplication of a spatial tuning kernel, $y(g(\theta, \varphi))$ and a temporal response profile $f(t - \tau)$:

$$Response(\theta, \varphi, t) = \left[ A \cdot \sum_{m=V,A,J,P} w_m \cdot f_m(t - \tau) \cdot y_m(g_m(\theta, \varphi)) + FR_0 \right] +$$

where A is the amplitude, $FR_0$ is spontaneous firing rate (baseline) and $w_m$ is the corresponding weight of each component. The [] + means that any fitted firing rate less than 0 would be set to 0. For models with different components, $f(t - \tau)$ can be defined as following with $\tau$ representing the delay time for neural response:

$$\text{Acceleration}: f_a(t - \tau) = \frac{(t - \tau)}{\sigma^2} \cdot e^{-\frac{(t-\tau)^2}{2 \cdot \sigma^2}}$$

$$\text{Velocity}: f_v(t - \tau) = e^{-\frac{(t-\tau)^2}{2 \cdot \sigma^2}}$$

$$\text{Position}: f_p(t - \tau) = \int_{-\infty}^{t} e^{-\frac{(t-\tau)^2}{2\sigma^2}} dt$$

$$\text{Jerk}: f_j(t - \tau) = -\frac{(t - \tau)^2 - \sigma^2}{\sigma^4} \cdot e^{-\frac{(t-\tau)^2}{2 \cdot \sigma^2}}$$

and in the spatial tuning component, $g(\theta, \varphi)$ is the cosine of the difference between each direction and the preferred direction, with $\theta$ and $\varphi$ indicating the preferred azimuth and elevation. $y(x)$ is a linear function that is typically used to describe non-cosine spatial tuning frequently seen in central

vestibular neurons instead of the peripheral afferents (*Angelaki and Dickman, 2000*). It is given by *Laurens et al., 2017*:

$$y(x) = o + (1 - |o|) \cdot x,$$

where *o* is the offset parameter, ranging from −1 to 1, allowing the spatial tuning to vary in both amplitude and width.

To evaluate model fit quality, we use the Bayesian Information Criterion (BIC) (*Schwarz, 1978*) that could calibrate extra fitting benefit from additional free parameters included in the model. BIC is given by:

$$\text{BIC} = n \cdot \ln\frac{RSS}{n} + p \cdot \ln n,$$

where RSS is the residual sum of squares, *n* is the number of data points and *p* is the number of free parameters.

To compare with other brain areas, we defined an index called the BIC difference index (BDI):

$$\text{BDI} = \frac{BIC_{VA\ model} - BIC_{PVAJ\ model}}{|BIC_{PVAJ\ model}|},$$

the data was fitted better by VA model if the BDI value is less than 0, and vice versa for the PVAJ model if the BDI value is greater than 0.

## Acknowledgements

We thank Wenyao Chen for monkey care and training, and Ying Liu for C++ software programming. This work was supported by grants from the National Natural Science Foundation of China Project (31761133014), the Strategic Priority Research Program of CAS (XDB32070000), and the Shanghai Municipal Science and Technology Major Project (2018SHZDZX05) to Y.G.

## Additional information

### Funding

| Funder | Grant reference number | Author |
| --- | --- | --- |
| National Natural Science Foundation of China | 31761133014 | Yong Gu |
| CAS | XDB32070000 | Yong Gu |
| Shanghai Municipal Science and Technology Commission | 2018SHZDZX05 | Yong Gu |

The funders had no role in study design, data collection and interpretation, or the decision to submit the work for publication.

### Author contributions

Bingyu Liu, Data curation, Formal analysis, Investigation, Methodology, Writing - original draft; Qingyang Tian, Data curation, Formal analysis; Yong Gu, Conceptualization, Resources, Data curation, Supervision, Methodology, Writing - original draft, Writing - review and editing

### Author ORCIDs

Yong Gu (ID) https://orcid.org/0000-0003-4437-8956

### Ethics

Animal experimentation: This study was performed in strict accordance with the recommendations in the Guide for the Care and Use of Laboratory Animals of Shanghai Institutes for Biological Sciences. All of the animals were handled according to approved institutional animal care and use committee (IACUC) protocols of the Shanghai Institutes for Biological Sciences. The protocol was

approved by the Committee on the Ethics of Animal Experiments of the Shanghai Institutes for Biological Sciences (Permit Number: ER-SIBS-221409P). All surgery was performed under sodium pentobarbital anesthesia, and every effort was made to minimize suffering.

### Decision letter and Author response
Decision letter https://doi.org/10.7554/eLife.64569.sa1
Author response https://doi.org/10.7554/eLife.64569.sa2

## Additional files

### Supplementary files
• Transparent reporting form

### Data availability
All data generated or analysed during this study are included in the manuscript and supporting files. Data https://doi.org/10.5061/dryad.xpnvx0kdd.

The following dataset was generated:

| Author(s) | Year | Dataset title | Dataset URL | Database and Identifier |
|---|---|---|---|---|
| Gu Y | 2021 | Date from: Robust vestibular self-motion signals in macaque posterior cingulate region | http://dx.doi.org/10.5061/dryad.xpnvx0kdd | Dryad Digital Repository, 10.5061/dryad.xpnvx0kdd |

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
