## [Decision Letter]

**Acceptance summary:**

This study provides a rigorous, systematic analysis of 3D translational and rotational self-motion signals in the posterior cingulate and retrosplenial cortex. Strengths of the study include the comparison across brain areas; the experimental design to distinguish the contribution of vestibular, visual and other cues to the neural responses; and the modeling approach used to analyze the data.

**Decision letter after peer review:**

Thank you for submitting your article "Robust vestibular self-motion signals in macaque posterior cingulate region" for consideration by *eLife*. Your article has been reviewed by 3 peer reviewers, including Jennifer L Raymond as the Reviewing Editor and Reviewer #1, and the evaluation has been overseen by Tirin Moore as the Senior Editor. The following individual involved in review of your submission has agreed to reveal their identity: Jean Laurens (Reviewer #2).

The reviewers have discussed the reviews with one another and the Reviewing Editor has drafted this decision to help you prepare a revised submission.

Summary:

This manuscript characterizes the vestibular signals carried by neurons in the posterior cingulate cortex and restrosplenial cortex. The PPC has long been implicated in conveying self-motion signals to the navigation system. However, previous work did not disentangle the contribution of various sensory modalities to this function. The present study uses controlled visual and vestibular stimuli to demonstrate a strong pattern of vestibular responses in PPC, with weaker self-motion signals in RSC. The experiments are systematic and rigorous, with appropriate controls and comparisons across conditions. The manuscript could be improved by several changes in the way the results are analyzed and presented to facilitate comparison of vestibular signaling across regions of the cortex, including previously published results. The work provides foundational knowledge of that should be of interest to scientists studying sensory coding as well as those interested in spatial navigation.

Essential revisions:

1. The key question is not whether the vestibular responses are "significantly better" fit with a PVAJ model compared with VA-only model (which is roughly what BIC measures), but how much the various components influence the neuron's responses. Partial coefficient of correlation would be better suited than BIC to describe how various components contribute to neuronal responses.

2. The reviewers found the analysis of temporal tuning to be confusing, and thought it would be better to omit that analysis and go straight to the VA model.

3. The rationale for using non-linear rather than linear spatial tuning functions was not clear. Using a linear function would have the dual advantages of reduced chance of overfitting and allowing comparison with published results from other cortical areas (Laurens et al., *eLife* 2017).

4. The model allowed varied delay between the different components; this choice was not well justified, and lines 550-552 of the Discussion indicate that the inclusion of this delay affected the results. Please elaborate on this in the Results section, providing information about the delays obtained in the model fits, and more adequate justification of the choice of a varied temporal delay.

5. The source of the data from the MSTd is not clear. Were these neurons also recorded for this study? Or if data was used from another study, it should be clearly stated which study and the methods for recording these data.

[Editors' note: further revisions were suggested prior to acceptance, as described below.]

Thank you for resubmitting your work entitled "Robust vestibular self-motion signals in macaque posterior cingulate region" for further consideration by *eLife*. Your revised article has been reviewed by 3 reviewers, one of whom is a member of our Board of Reviewing Editors, and the evaluation has been overseen by Tirin Moore as the Senior Editor.

Summary:

The point-by-point response indicates that the authors seriously considered the input from the reviewers. Unfortunately, some of the major concerns raised remain largely unresolved.

The manuscript has been improved but there are some remaining issues that need to be addressed, as outlined below:

1. Better address the concerns raised about the inclusion of a variable and lengthy delay for each signal in the model, and assumptions about which delays are longer

2. Do a more complete partial correlation analysis

3. Address concerns about overfitting

Reviewer #1:

The point-by-point response indicates that the authors seriously considered the input from the reviewers. Unfortunately, two of the major concerns raised remain largely unresolved.

I am particularly concerned about the inclusion of variable delays for each temporal component in the model. The range of relative delays between V and A components is quite broad, with many fits up to 300 ms. The authors describe that these delays are "reasonable", but without justification, and I am not persuaded that delay is reasonable for a vestibular signal, even at higher levels in cortical processing. Moreover, it is not clear why the figures show the relative delays between temporal components, rather than the absolute delay fit by the model for each component-P,V,A,J, and I am worried that those delays may be even longer and less plausible. Notably, the delays for the VA vs PVAJ models are different, with many shorter delays for the latter, suggested something is amiss with this element of the model, particularly in the VA model. Finally, in the supplementary figures showing results when a fixed delay was used (Figure 8-Supp 1 and Figure 10-Supp2), the legends do not make it clear that is what is being shown, or what the fixed delay was.

The rationale provided in the manuscript for using nonlinear rather than linear fits has not been extended at all. The point-by-point response indicates that the authors tried the linear model. It would be instructive to include in the manuscript documentation of shortcomings of the linear model, and what is gained by adding the nonlinearity.

Reviewer #2:

In this revision, the authors have addressed our comments, but with varying degree of success. For instance:

– The partial correlation analysis performed in response to comment 1 is not based on 3D model fits. Instead, they performed separate analysis is based on only 1 motion direction (which won't represent the contribution of different components accurately if their spatial tuning are not aligned). Furthermore, this analysis doesn't take the delays of various components into account. Furthermore, this analysis is presented in a very superficial manner (Figure 10S1). Overall, they performed an un-insightful partial correlation analysis and then added two main figures to present the PVAJ model: this makes the manuscript more complicated for little benefit.

– The authors didn't really shorten or eliminate the analysis of temporal tuning (as suggested in comment 2 to streamline the manuscript) but merely moved ten lines to Methods.

– Regarding the delays between various components (comment 4), they added histograms of these delays in Figure 8D and Figure 10E. However, these histograms show that these delays are always positive, e.g. they assumed that V always lags A (the axis label in these panels is ambiguous, btw). This assumption is justified in the text as follows: "the velocity time was set to lag the acceleration time because vestibular velocity signal is supposed to be an integral from the acceleration quantity" (lines 289-291).

Unfortunately, this is mathematically incorrect: first, just because velocity is the integral of acceleration doesn't imply that it "lags" it (except in special cases e.g. using sinusoids); those are completely different notions. Second, even if velocity did lag acceleration, this fact would already be represented by the shape of the temporal components used to fit the model, and imposing a positive delay between acceleration and velocity would still be incorrect. Likewise, the authors assume that position lags velocity for the same incorrect reason (lines 372-374) and that jerk lags acceleration with a justification which is quite incomprehensible (lines 371-372).

In principle, I can appreciate that having variable delays may be justified; for instance if different neuronal pathways conveying different dynamic components converge onto a cortical neuron. On the other hand, I have reservations about this method as it may promote overfitting. In any case, if the authors introduce variable delays, then they must allow each delay to be positive or negative since there is no justification for doing otherwise.

– On the other hand, they addressed a number of other issues correctly.

Overall, I find the current revision a little disappointing: the manuscript has become more complicated (two new figures) and, even though some issues were corrected, new were raised. Yet, I think that this work may be important enough to consider publication if these issues can be addressed in a new round of corrections.

Reviewer #3:

I think the authors have addressed the majority of points raised by the reviewers quite well.

They expanded their PVAJ to VA model comparison by partial correlation analysis, confirming the results they initially saw by only using BIC.

The reworked figures are better to read and understand and the data is a lot clearer in its representation.

The descriptions are clearer now and missing parts were added to the method section.

However, the main text still needs some editing in terms of language/grammar.

Examples:

line 171 about half neurons – about half of the neurons

Line 226 instead from other cues – instead of from other cues

Regarding the fixed versus varied lag:

It should be added to the figure legends when it shows fixed lag. Also, from visual inspection, the fixed delay does not necessarily look like a worse fit (at least for the PVAJ model, Figure 10D and Figure10—figure supplement 2), how was this quantified?

---

## [Author Response]

Essential revisions:1. The key question is not whether the vestibular responses are "significantly better" fit with a PVAJ model compared with VA-only model (which is roughly what BIC measures), but how much the various components influence the neuron's responses. Partial coefficient of correlation would be better suited than BIC to describe how various components contribute to neuronal responses.

We agree with the reviewer’s comment about the key question which is “how much the various components influence the neuron's responses”. We took the reviewer’s suggestion by performing the partial correlation analyses. Specifically, we computed partial correlation coefficients of the neural responses with each temporal component for each cell (i.e., velocity, acceleration, jerk and position, see method at line 822-828). As shown in the new Figure 10—figure supplement 1, partial correlation analysis nicely revealed many significant cases for each temporal component, with position relatively weaker compared to the others, confirming that these components are important. However, these coefficients tend to be high, and except for the position signal, the difference among the other temporal components is small and hard to be compared. So, in addition to the partial correlation analysis, we further performed the 3D spatial-temporal model (PVAJ) analysis in a similar way as developed in previous study (Laurens et al., 2017). This method not only gives the weight of each temporal component for each cell, reflecting their relative contributions, but also consider temporal effects in each spatial direction, allowing to provide more detailed and thorough information about spatial-temporal relationship. We have now modified texts accordingly, and these new results were added in the main text (line 358-368), method (line 822-828), and Figure 10—figure supplement 1.

2. The reviewers found the analysis of temporal tuning to be confusing, and thought it would be better to omit that analysis and go straight to the VA model.

We apologize for the confusion caused by our description. In our experiment, to be able to assess the overall tuning situations in the targeted areas, we have performed unbiased recording by including every neuron we encountered online once we were certain to be in the right area. We then used the criterion used by previous researchers (e.g., Chen et al., 2010) to select neurons with significant temporal tuning for further analysis. Thus, the analysis of temporal tuning is traditional, yet it is pretty important for most of the consequent analyses in the current study. But we agree with the reviewer that our previous description is problematic, so we have now heavily polished this part, by moving large part of the detail to the Method to clearly explain this operation (line 775-787). By contrast, we only keep a very brief and clear description in the main text to affiliate reading (line 166-168). We hope this adjustment would be satisfactory, but we could further make modifications if reviewers still have concerns.

3. The rationale for using non-linear rather than linear spatial tuning functions was not clear. Using a linear function would have the dual advantages of reduced chance of overfitting and allowing comparison with published results from other cortical areas (Laurens et al., eLife 2017).

The reviewer is correct in that the linear function in Laurens’ paper is potentially better in reducing overfitting and allowing comparison with results from the published results. We originally simply chose the nonlinear function from a previous paper (Chen et al., 2011a) because we found this function can explain our data well and make sense theoretically. But we have not thought about the reviewer’s point about its alternatives. Sorry for this negligence. We have now tried the linear spatial function in Laurens’ paper, and found pretty similar results in general. However, the spatial congruency among different temporal components became less consistent. Moreover, because we have used a varied-delay setting that has not been used by previous studies anyway (see reply-to-reviewer point #4), we still keep the nonlinear function in the current version after careful weighing. We hope that this is acceptable, yet please let us know if the reviewer still has the concern.

4. The model allowed varied delay between the different components; this choice was not well justified, and lines 550-552 of the Discussion indicate that the inclusion of this delay affected the results. Please elaborate on this in the Results section, providing information about the delays obtained in the model fits, and more adequate justification of the choice of a varied temporal delay.

We apologize for the vague information. There are a number of reasons we decided to include a varied delay. One is that it is a bit hard for us to suppose that all neurons have identical delays between each temporal component. Instead, it is likely that these components could be inherited from upstream neurons at different hierarchical levels, leading to heterogeneous lags. The other reason is that we see more consistent spatial information among different temporal components, compared to the case when the delay is fixed. This applies to both of the VA model and the full PVAJ model. We argue this result may make more sense in terms of efficiency of sensory coding, yet we don’t have further stronger evidence in the current study to support it, which definitely requires future studies. We have discussed this in the Discussion in the previous version, but now elaborate on this in the Results section (line 290-296, 343-346, 371-377). Following the reviewers, we also added the distribution of different temporal delays in the new Figure 8 and Figure 10 for VA model and PVAJ model, respectively. As a comparison, spatial correlation based on fixed-delay as used in previous models were presented in the Figure 8—figure supplement 1 and Figure 10—figure supplement 2.

5. The source of the data from the MSTd is not clear. Were these neurons also recorded for this study? Or if data was used from another study, it should be clearly stated which study and the methods for recording these data.

We apologize for the missing information. MSTd data were from another study (Gu et al., 2006). We now have added the detail in the Method (line 766-767).

[Editors' note: further revisions were suggested prior to acceptance, as described below.]

The manuscript has been improved but there are some remaining issues that need to be addressed, as outlined below:1. Better address the concerns raised about the inclusion of a variable and lengthy delay for each signal in the model, and assumptions about which delays are longer

We seriously reconsidered the reviewers’ suggestions and concerns about the varied delay assumed in the previous version of our spatiotemporal model. We do agree with the reviewers that including the varied delay would potentially introduce more parameters and subsequently cause overfitting. Plus, as the reviewer pointed out, the assumptions about the relative delay among each temporal component is also kinda arbitrary. We could not really argue with persuasive evidence to show that our previous model really gains more benefits over the ones the reviewers proposed. Thus, we do not insist on our own model anymore, and decided to switch to the one with fixed-delay to eliminate issues as raised from all the reviewers.

Related with this issue, we also agree with the reviewer that the linear function is truly better than the nonlinear one to reduce potential overfitting. Thus in sum, we now used a model with fixed-delay (i.e., same delay time for all temporal components) plus a linear function (Laurens et al., 2017) in the new version of the manuscript. Importantly, we show that this decision does not change our main conclusions in our study. Hence, we hope that this modification would be acceptable for all the reviewers.

2. Do a more complete partial correlation analysis

We apologize for the previous inappropriate partial correlation analysis. We now have performed the analysis in a more correct way. In particular, we now calculated the partial correlation coefficients between neuronal responses (PSTHs) and the fitted data in all motion directions (26 vectors) for each temporal component in the 3D spatiotemporal model. As shown in the new Figure 10B, partial coefficients of correlation indicate significant contributions from all 4 temporal components for majority of the neurons in PCC. We now have included this result in the main text (line 345-351, & Figure 10B).

3. Address concerns about overfitting

Please see the reply to Question 1. Briefly, since we now have reduced the number of parameters by switching to a fixed-delay model with linear functions as suggested by the reviewers, we believe that this change would largely release the concerns about overfitting.

Reviewer #1:The point-by-point response indicates that the authors seriously considered the input from the reviewers. Unfortunately, two of the major concerns raised remain largely unresolved.I am particularly concerned about the inclusion of variable delays for each temporal component in the model. The range of relative delays between V and A components is quite broad, with many fits up to 300 ms. The authors describe that these delays are "reasonable", but without justification, and I am not persuaded that delay is reasonable for a vestibular signal, even at higher levels in cortical processing. Moreover, it is not clear why the figures show the relative delays between temporal components, rather than the absolute delay fit by the model for each component-P,V,A,J, and I am worried that those delays may be even longer and less plausible. Notably, the delays for the VA vs PVAJ models are different, with many shorter delays for the latter, suggested something is amiss with this element of the model, particularly in the VA model. Finally, in the supplementary figures showing results when a fixed delay was used (Figure 8-Supp 1 and Figure 10-Supp2), the legends do not make it clear that is what is being shown, or what the fixed delay was.The rationale provided in the manuscript for using nonlinear rather than linear fits has not been extended at all. The point-by-point response indicates that the authors tried the linear model. It would be instructive to include in the manuscript documentation of shortcomings of the linear model, and what is gained by adding the nonlinearity.

We agree with the reviewer’s concerns about the varied-delay and nonlinear function used in the previous version of model. We have now switched it to the fixed-delay plus the linear function as used in previous study (Laurens, et al., 2017). This switch does not change the main conclusion of our study, so we hope this modification is now satisfactory to the reviewer. Please see our common reply above as well.

Reviewer #2:In this revision, the authors have addressed our comments, but with varying degree of success. For instance:– The partial correlation analysis performed in response to comment 1 is not based on 3D model fits. Instead, they performed separate analysis is based on only 1 motion direction (which won't represent the contribution of different components accurately if their spatial tuning are not aligned). Furthermore, this analysis doesn't take the delays of various components into account. Furthermore, this analysis is presented in a very superficial manner (Figure 10S1). Overall, they performed an un-insightful partial correlation analysis and then added two main figures to present the PVAJ model: this makes the manuscript more complicated for little benefit.

We apologize for the previous partial correlation analysis that was not performed appropriately. We now have performed the analysis in a more correct way. In particular, we now calculated the partial correlation coefficients between neuronal responses (PSTHs) and the fitted data in all motion directions (26 vectors) for each temporal component in the 3D spatiotemporal model. As shown in the new Figure 10B, the partial coefficients of correlation indicate significant contributions from all 4 temporal components for majority of the neurons in PCC. Plus, since we have now switched to a fixed-delay model, so the issue of varied-delay should not be there anymore. We now have included this result in the main text (line 345-351), as well as changing the figures (Remove the old ones and added the new Figure 10B).

– The authors didn't really shorten or eliminate the analysis of temporal tuning (as suggested in comment 2 to streamline the manuscript) but merely moved ten lines to Methods.

We now have further shortened the analysis of temporal tuning and the corresponding texts a lot. We hope this modification is satisfactory now.

– Regarding the delays between various components (comment 4), they added histograms of these delays in Figure 8D and Figure 10E. However, these histograms show that these delays are always positive, e.g. they assumed that V always lags A (the axis label in these panels is ambiguous, btw). This assumption is justified in the text as follows: "the velocity time was set to lag the acceleration time because vestibular velocity signal is supposed to be an integral from the acceleration quantity" (lines 289-291).Unfortunately, this is mathematically incorrect: first, just because velocity is the integral of acceleration doesn't imply that it "lags" it (except in special cases e.g. using sinusoids); those are completely different notions. Second, even if velocity did lag acceleration, this fact would already be represented by the shape of the temporal components used to fit the model, and imposing a positive delay between acceleration and velocity would still be incorrect. Likewise, the authors assume that position lags velocity for the same incorrect reason (lines 372-374) and that jerk lags acceleration with a justification which is quite incomprehensible (lines 371-372).In principle, I can appreciate that having variable delays may be justified; for instance if different neuronal pathways conveying different dynamic components converge onto a cortical neuron. On the other hand, I have reservations about this method as it may promote overfitting. In any case, if the authors introduce variable delays, then they must allow each delay to be positive or negative since there is no justification for doing otherwise.

We appreciate the reviewer’s comment and agree with the reviewer’s concerns about the varied-delay. We do realize that the varied-delay assumptions, particularly about the relative delay among each temporal component is kinda arbitrary and incorrect. Since this is a general issue raised by all the reviewers, we now decided to eliminate this setting by switching to the fixed-delay model. Please also see our reply to the common Question 1.

Reviewer #3:I think the authors have addressed the majority of points raised by the reviewers quite well.They expanded their PVAJ to VA model comparison by partial correlation analysis, confirming the results they initially saw by only using BIC.The reworked figures are better to read and understand and the data is a lot clearer in its representation.The descriptions are clearer now and missing parts were added to the method section.However, the main text still needs some editing in terms of language/grammar.Examples:line 171 about half neurons – about half of the neuronsLine 226 instead from other cues – instead of from other cues

Thank you for pointing these problems out. We now have corrected them.

Regarding the fixed versus varied lag:It should be added to the figure legends when it shows fixed lag. Also, from visual inspection, the fixed delay does not necessarily look like a worse fit (at least for the PVAJ model, Figure 10D and Figure 10—figure supplement 2), how was this quantified?

Since the varied-delay is a general issue raised by all the reviewers, we now decided to eliminate this setting by switching to the fixed-delay model. The main conclusion of our current work does not change. We hope this modification is satisfactory. Please also see our common reply above as well.